# Targeted computational design of an interleukin-7 superkine with enhanced folding efficiency and immunotherapeutic efficacy

See-Khai Lim[1], Wen-Ching Lin[1], Yi-Chung Pan[1], Sin-Wei Huang[1], Yao-An Yu[1], Cheng-Hung Chang[1], Che-Ming Jack Hu[1]\*, Chung-Yuan Mou[2]\*, Kurt Yun Mou[1]†

[1]Institute of Biomedical Sciences, Academia Sinica, Taipei, Taiwan; [2]Department of Chemistry, National Taiwan University, Taipei, Taiwan

## eLife Assessment

This **important** study presents the rational redesign and engineering of interleukin-7. The data from the integrated approach of using computational, biophysical, and cellular experiments are **convincing**. This paper is broadly relevant to those studying immunomodulation using biologics.

**\*For correspondence:**
chu@ibms.sinica.edu.tw (C-MJH);
cymou@ntu.edu.tw (C-YM)

†Deceased

**Abstract** Interleukin-7 (IL-7) plays a central role in maintaining T cell development and immune homeostasis, and enhancing the cytokine's immune-stimulatory functionality has broad therapeutic implications against various oncological malignancies. Herein, we show a computationally designed IL7 superkine, Neo-7, which exhibits enhanced folding efficiency and superior binding affinity to its cognate receptors. To streamline the protein candidate prediction and validation process, the loop region of IL7 was strategically targeted for redesign while most of the receptor-interacting regions were preserved. Leveraging advanced computational tools such as AlphaFold2, we show loop remodeling to rectify structural irregularities that allow for iterative stabilization of protein backbone and lead to identification of beneficial mutations conducive to receptor engagement. Neo-7 superkine shows improved thermostability and production yield, and it exhibits heightened immune-stimulatory and anticancer effect in C57BL/6 J mice. Neo-7 addresses intrinsic developability limitations of IL-7, including inefficient folding, aggregation propensity, and suboptimal receptor engagement, while in vivo pharmacokinetic limitations of wild-type IL-7 were addressed separately through Fc fusion. These findings underscore the utility of a targeted computational approach for de novo cytokine development.

## Introduction

Clinical success with immune checkpoint inhibitors (ICIs) and immunogenic cell death (ICD) inducers has invigorated developmental efforts on immunotherapeutic strategies that strengthen the body's immune system against oncological malignancies (*Lim et al., 2024*; *Galluzzi et al., 2024*; *Waldman et al., 2020*; *Tan et al., 2020*; *Tan et al., 2020*; *Taefehshokr et al., 2022*). The promise of durable tumor suppression by immunotherapeutic approaches has rekindled enthusiasm towards cytokine therapies, which can enhance immune responses and promote a more effective attack on cancer cells by directly stimulating the proliferation and activation of various immune cell types (*Park et al., 2024*; *Fu et al., 2023*; *Waickman et al., 2016*). Cytokine-mediated immune cell activation can override the negative feedback mechanisms and bolster immune cell stimulation, allowing for broad-spectrum

tumor suppression and synergistic actions with ICIs and ICD inducers (*Holcomb and Zou, 2022*; *Shen et al., 2020*). However, traditional cytokine therapies have been hampered by challenges such as short half-lives, poor pharmacokinetics, and significant toxicity at therapeutic doses (*Fu et al., 2023*; *Silva et al., 2019*). These issues have hindered the broader application of cytokines in cancer treatment despite their potential benefits. For instance, IL-2, one of the first cytokines identified with potent anti-cancer properties, is limited by severe adverse effects, restricting its maximum tolerated dose (MTD) to 37 µg/kg, which in turn limits its efficacy (*Mizui, 2019*). Similar toxicity concerns exist for other interleukins, such as IL-12, IL-15, and IL-21, with clinical MTDs of 1.0, 20, and 100 µg/kg, respectively (*Knudson et al., 2020*; *Steele et al., 2012*; *Portielje et al., 1999*), thus prompting emerging protein design strategies for cytokine variant development with elevated efficacy profiles.

Advances in computational protein design have opened up new opportunities for cytokine engineering, which enables de novo design of cytokine variants with improved stability, reduced toxicity, and enhanced efficacy. These engineered cytokines can be structurally modified for enhanced engagement with their putative cognate receptors, leading to enhanced therapeutic and safety profiles for treatment development. A notable example is the work by *Silva et al., 2019*, which demonstrated a significant breakthrough in computational protein design (*Silva et al., 2019*). Using the Rosetta protein design suite, they restructured and redesigned four-helix bundle mimetics of IL-2 and IL-15 (neo-2/15), resulting in proteins that are more potent and stable both in vitro and in vivo, with specifically programmed receptor specificity. Although the seminal study showcased the possibility for researchers to design novel protein with computationally idealized structure in its entirety, such approach demands substantial computing resources and can generate many non-analogous candidates for validation. We envision that a targeted redesign strategy that focuses on protein domains with manifest folding inefficiency can minimize the resources required for computational protein engineering. In the present work, the targeted computational design strategy is demonstrated through the development of an IL-7 variant.

Interleukin-7 (IL-7) is a member of the common gamma chain (γc) receptor family, which includes cytokines such as IL-2, IL-15, and IL-21 (*Waickman et al., 2016*; *Lin and Leonard, 2018*; *Sim and Radvanyi, 2014*). IL-7 plays a crucial role in the survival, differentiation, and proliferation of immune cells, including T-lymphocytes, B-lymphocytes, and natural killer (NK) cells (*Park et al., 2024*; *Winer et al., 2022*). In vitro studies have shown that IL-7 promotes the survival and homeostatic proliferation of splenocytes by upregulating survival factors like Bcl-2. Preclinical animal studies and clinical trials have demonstrated broader functions, including the induction and maintenance of memory T cells, promotion of T cell homing to lymphoid organs, increased T cell receptor diversity, and enhanced T cell cytotoxicity by the cytokine (*Park et al., 2024*; *Winer et al., 2022*; *Lee et al., 2024*; *Kim et al., 2020*). The first-generation IL-7, known as CYT-107, demonstrated potent immune cell reconstitution in phase 2 clinical trials for HIV patients (for CD4+ T cell restoration) and patients with sepsis-induced lymphopenia (*Trédan et al., 2018*; *Perales et al., 2012*; *Sportès et al., 2010*). However, its anti-cancer efficacy was limited, as it increased immune cell count but did not induce a significant anti-tumor response, likely due to its short half-life (8.7–34.6 hr; *Park et al., 2024*; *Winer et al., 2022*; *Trédan et al., 2018*; *Fu et al., 2024*). The second-generation IL-7 drug, NT-I7 (efineptakin alfa), uses a hybrid human IgG/IgD Fc domain for half-life extension (T1/2=60.8–139.7 hr), which significantly improves its anti-tumor activities. Efineptakin alfa supports the proliferation and survival of CD4+ and CD8+ cells in both humans and mice, showing promising responses in early-phase clinical trials for advanced solid tumors, particularly in combination with immune checkpoint inhibitors (*Park et al., 2024*; *Kim et al., 2020*; *Fu et al., 2024*). Although much effort has been devoted to enhancing IL-7 pharmacokinetics through protein fusion, no variant or computationally designed mimic has been reported on the cytokine to our knowledge.

From a development perspective, IL-7 is prone to aggregation and requires additional engineering to enhance its solubility and stability. While prior clinical efforts have primarily addressed IL-7's short circulating half-life through fusion strategies such as Fc conjugation, comparatively less attention has been paid to intrinsic scaffold inefficiencies of IL-7, including long inter-helical loops, aggregation propensity, and folding dependence on multiple disulfide bonds. To address these challenges, we herein engineer a novel variant of IL-7, known as Neo-7, with enhanced folding efficiency through targeted computational protein design. By removing redundant loops and redesigning the protein backbone using Rosetta loop remodeling, we reconnected the functional domains in a more efficient

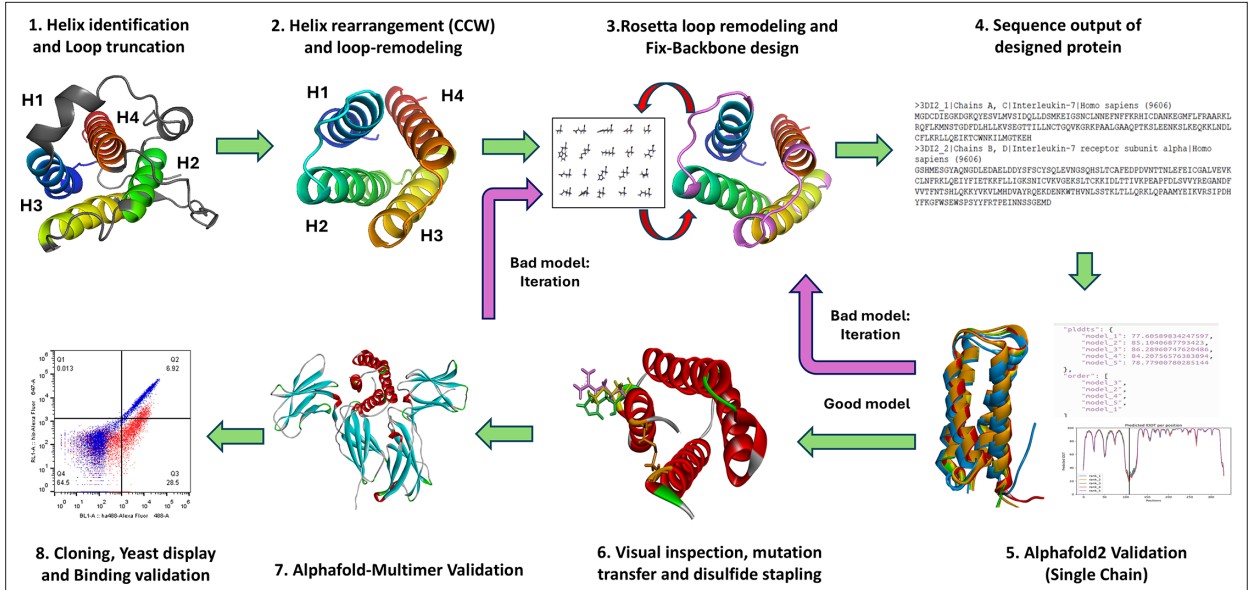

**Figure 1.** Schematic illustrating the computational pipelines for Neo-7 engineering from wild type IL7 sequence. In general, non-receptor interacting loops were deleted from the WT-IL7 sequence and loops connecting the adjacent helices were modeled using Rosetta Loop Remodeler and Rosetta fix backbone design function. The sequence of the designed model was extracted and submitted to AlphaFold (monomer and multimer mode for structure and protein-receptor binding prediction respectively) as a preliminary validation of the Rosetta-remodeled protein. Iterations of the bad models (models that do not fold into the expected structure or models that did not predict to bind to the receptors) back to the design stage were performed. Models that passed the AlphaFold validation proceeded to subsequent in vitro assay using yeast display system and flow cytometry to determine their relative binding affinity to IL-7 receptors in comparison to WT-IL7.

and economical structural (defined here as minimizing the number of residues required to connect receptor-binding or structural helices). Advanced computational tools were employed to ensure that the Neo-7 variant would fold according to the predicted structural models, resulting in a protein with superior folding efficiency and receptor binding affinity. In further experimental studies, Neo-7 demonstrates improved in vitro production yield and in vivo biological activity, positioning it as a promising candidate for cancer immunotherapy. These advancements underscore the potential of IL-7 and its engineered variants as effective cancer immunotherapies, highlighting the importance of optimizing cytokine properties for clinical application.

## Results

### Development of a computational pipeline for rational protein design

In redesigning IL-7, we aimed to minimize sequence alterations relative to the wild-type protein. This strategy is based on the understanding that a protein's sequence is closely tied to its structure and function. By preserving the native sequence elements involved in IL-7's biological activity, we sought to maximize the likelihood of producing a successful variant without compromising its original function. We began by examining the crystal structure of IL-7 bound to its receptor, IL7Rα (interleukin-7 receptor alpha; PDB ID: 3DI2), which recruits IL-2Rγ to form a heterodimeric receptor complex essential for downstream signaling. Loop remodeling was performed using Rosetta's loop remodeling modules, followed by validation with AlphaFold, employing both single-sequence monomer and multimer models. During the computational design and validation process, each model predicted by either Rosetta or AlphaFold was meticulously inspected using PyMOL and Discovery Studio 4.5. Residues that were found to clash with neighboring amino acids were further refined through mutations using Rosetta's fix backbone modules. The gene sequences of the final designs that successfully passed this computational pre-filtering were then synthesized and cloned into a yeast display vector for subsequent in vitro validation (*Figure 1*).

The wild-type IL-7 protein forms a 4-helix bundle structure with three inter-helical loops. Upon visually examining its crystal structure in complex with IL-7 receptor alpha (IL7Rα), we noted that

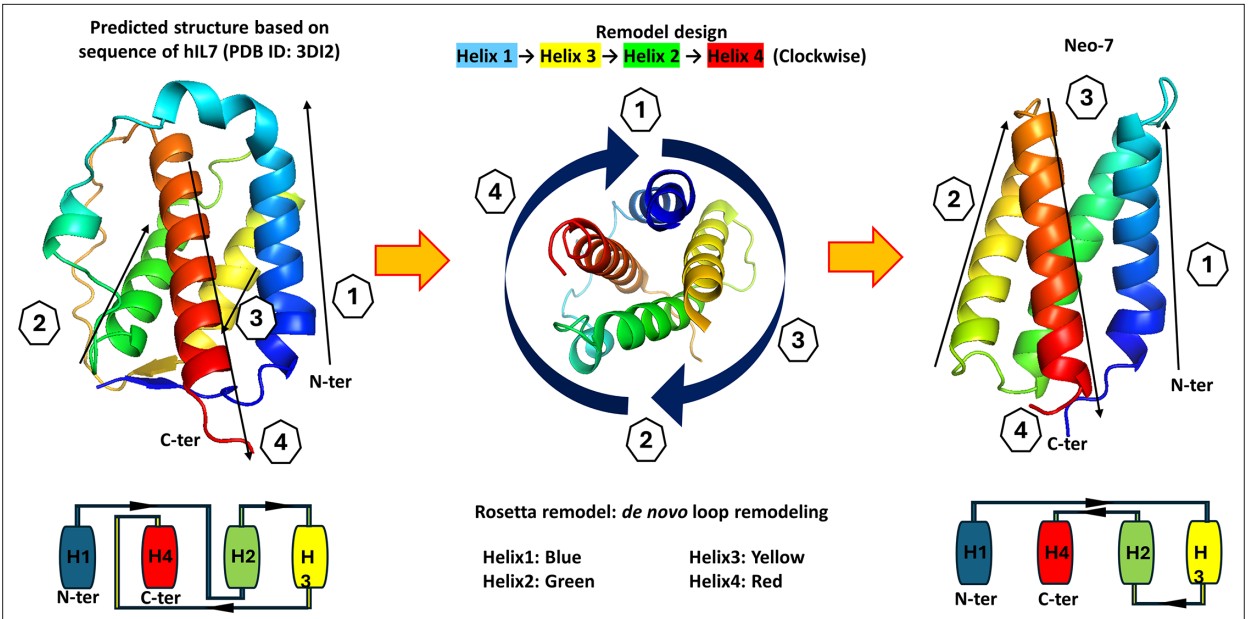

**Figure 2.** Blueprint of Neo-7 design. Blueprint of the WT-IL7 was shown on the left of the figure. The connectivity of the functioning helixes was connected in a manner that requires extremely long protein loops by design (i.e. helices were not connected to the closest adjacent helices but to the opposite helix). Loops that were not interacted with the IL-7 receptors were deleted and the helixes were reconnected in a clockwise manner via new protein linkers connecting to the adjacent helixes. The blueprint of the redesigned protein was shown at the right side of the figure. Protein structures are colored as rainbow (from N-to-C terminus with the order of Blue-Green-Yellow-Red).

the connectivity between each helix is intricate and inefficient (*Figure 2*). In the crystal structure of WT-IL7 complexed with IL-7 receptor alpha, IL-7 is composed of four helices. Helices 1 and 3 are found to interact with IL-7 receptor alpha (IL7Rα), while helices 1 and 4 are predicted to interact with the IL-2 receptor common gamma chain (IL2Rγ; *McElroy et al., 2009*). Helix 2 was observed to serve as a structural support for the helix bundle. However, the connectivity of these helices exhibited complexity and inefficiency. Notably, the loop between helix 1 and 2 spans 29 amino acids (AA), the loop connecting helix 2 and 3 is 6 AA long, and the loop connecting helix 3 and 4 is 35 AA long. We extracted the helices from the PDB structure file and reconnected the helices using the Loop Remodel function from the Rosetta suite. The protein was remodeled clockwise in the order of H1>H3>H2>H4. The new loops connecting H1-H3, H3-H2, and H2-H4 are 7, 8, and 5 AA in length, respectively. The amino acid sequences of the designed loops were subsequently optimized using Rosetta Fix Backbone design function and validated using Alphafold monomer.

### Structural modeling of Neo-7 in AlphaFold single sequence mode

The synthesized Neo-7 sequence was cloned into a yeast-display vector to assess its expression and binding capability to IL7Rα. Despite the concordance between Rosetta and AlphaFold monomer predictions, no binding was observed between displayed Neo-7 and IL7Rα (*Figure 3C*). A degree of binding to IL2Rγ was detected, possibly reflecting partial folding of the displayed protein in the yeast display platform. We also postulated that the high similarity between the helices of Neo-7 and WT-IL7 amino acid sequences might lead to sampling bias, leading AlphaFold to predict a similar folding of Neo-7 to WT-IL7. To investigate this, we utilized AlphaFold Single Sequence (AlphaFold SS), an online platform capable of performing AlphaFold without relying on a PDB template and multiple sequence alignment (MSA) database. The resulting structure model from AlphaFold SS indicated misfolding of helix 2 (providing structural support) and helix 3 (binding to IL7Rα) when superimposed onto WT-IL7 helices (*Figure 3A*). However, the helices responsible for binding to the gamma receptor aligned well with the WT template. Subsequently, we addressed the misfolded helices by redesigning their connecting loops (loop 1 and 2) using Rosetta fix backbone design and validated the structures with AlphaFold SS (*Figure 3B*).

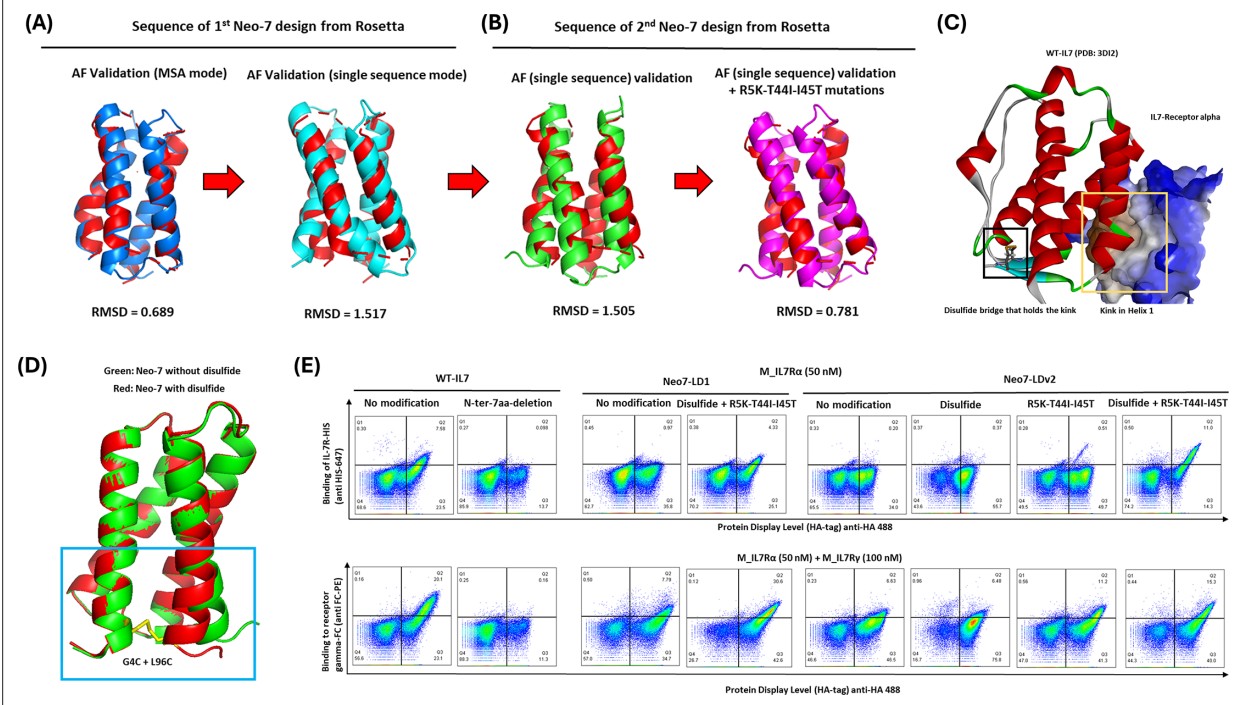

**Figure 3.** Validation of Neo-7 designs from Rosetta loop remodeling and fix backbone design. (**A**) AlphaFold validation of the first loop design version of Neo-7 (Neo-7 LDv1) using the default (left) and single sequence mode (right). (**B**) AlphaFold validation of the second loop design version of Neo-7 (left; Neo-7 LDv2) and Neo-7 LDv2 with mutations (right) favored by Rosetta fix backbone design. (**C**) Crystal structure of human IL-7 in complexation to human IL-7 receptor alpha (PDB ID = 3DI2). (**D**) Superimposition of Neo-7 structures (with or without additional disulfide bridge) predicted by AlphaFold. (**E**) Yeast display and flow cytometry validation of IL-7/Neo-7 bindings towards the IL-7 receptors. The yeast-displayed protein (different redesigned IL-7s) carries a HA-tag while the recombinant IL-7 receptors carry either a HIS tag (IL-7 receptor alpha) or a FC-tag (common-IL-2 family receptor gamma; for detection of IL-2Rγ binding, yeast cells were first incubated with recombinant IL-7Rα, washed, and subsequently incubated with IL-2Rγ.) The signal intensity of the X-axis (conferred by the binding of anti-HA mab) correlates with the expression level of the displayed protein while the signal intensity of the Y-axis (conferred by the binding of the anti-HIS/anti-FC mAb to the recombinant receptors bound to the displayed proteins) correlates with the binding affinity of the displayed proteins towards the IL-7 receptors.

The online version of this article includes the following figure supplement(s) for figure 3:

**Figure supplement 1.** Yeast cell flow cytometry gating strategy.

## Disulfide stapling on Helix 1 and stabilizing mutations are essential for binding of IL-7 and Neo-7 to IL-7 receptor-α

Since the helices were adapted from the WT-IL7, we hypothesized that those beneficial mutations to stabilize IL-7 or enhance its binding to the cognate receptors should be applicable to our designed constructs. Prior to the transfer of the beneficial mutations discovered from our previous study (*Figure 3—figure supplement 1*), we first constructed a multimer model consisting of Neo-7, murine IL7Rα, and the common IL2Rγ receptors using AlphaFold Multimer. The resulting model was used as a structure template for Rosetta fix backbone design, which the amino acid variants of the mutants and their wild-type counterpart (*Table 1*) were keyed into the Rosetta fix backbone residue files for

**Table 1.** Site-directed mutagenesis residues recommended by Rosetta to improve the folding stability of the Neo-7s.

| Residue number | Wild type | Mutant | Remarks | Rosetta's choice |
|---|---|---|---|---|
| 5 | K | R | Cytokine-receptor binding interface | R |
| 6 | Q | P | Cytokine-receptor binding interface | Q |
| 44 | T | I | Cytokine core-surface interface | I |
| 45 | I | T | Borderline of cytokine binding interface | T |

calculation. Only those mutations that were favored by Rosetta are selected to maximize the stability of the protein and to preserve their theoretical folding in the subsequent in vitro yeast display assay.

The Neo-7 design, incorporating Rosetta-favored mutations, demonstrated enhanced protein-display level and binding to IL2Rγ (*Figure 3E*). However, despite the alignment between computational protein design and prediction programs, weak or insignificant binding of the designed Neo-7 to murine IL7Rα was observed in the yeast display system. A closer examination of the crystal structure of WT-IL7 with IL7Rα revealed a kink on helix 1 is crucial for proper IL7-IL7Rα binding, stabilized by a disulfide bridge on the N-terminus and loop 3 of WT-IL7 (*Figure 3C*). The structure of Neo-7 generated from AlphaFold lacked this kink, possibly due to the absence of the disulfide bridge. To address this, we hypothesized that regenerating the kink through mutations of core amino acids between helix 1 and 4 into cysteine might recapitulate the native kink conformations observed in WT-IL7 (*Figure 3C*). Introducing G4C and L96C mutations into Neo-7 validated this hypothesis, as the resulting disulfide-stapled Neo-7 model predicted by AlphaFold SS exhibited the reappeared kink on helix 1 (*Figure 3D*). Encouraged by this result, we proceeded to the next stage of in vitro evaluation via yeast display using the designed Neo-7.

Meanwhile, to assess the importance of the kink for IL7Rα binding affinity in WT-IL7, we deleted the first 7 amino acids at the N-terminus of WT-IL7 and investigated its binding to murine IL7Rα in the yeast display system. Interestingly, the results (*Figure 3E*) demonstrated that N-terminal truncation fully diminished the binding of IL-7 to IL7Rα. Conversely, for the Neo-7 design, both the disulfide stapling and Rosetta-favored mutations were found essential for binding to IL7Rα. This indicates that both the helix kink from the disulfide design and core stability conferred by the transferred mutations are required for the generation of a stable and functional Neo-7. Importantly, the enhanced binding affinity towards IL7Rα also led to improved binding towards the common IL2Rγ, relative to other variants in the Neo-7 series.

## Transfer of beneficial mutations on the cytokine-receptor binding interface further enhanced the binding of Neo-7 to IL7-Receptor α

Encouraged by the successful expression of a functional Neo-7 (Neo-7 Loop design 2 with R5K-T44I-I45T mutations, herein referred to as Neo-7) design in the yeast display platform, we modeled the structure of Neo-7 in the presence of murine IL7Rα and IL2Rγ. After careful examination of all the mutation points (*Figure 4A*), we hypothesized that two mutation points might further improve the binding affinity of Neo-7 towards IL7Rα due to the following reasons: (1) Both residues were located closely to a hydrophobic patch of IL7Rα and replacing the polar glutamine and threonine into a more hydrophobic proline or isoleucine is favorable for binding through hydrophobic interactions (2) Proline was disfavored by Rosetta probably due to its specific characteristics as a helix breaker (*Table 1*). However, after careful inspection of the structure, we observed that proline was located at the starting

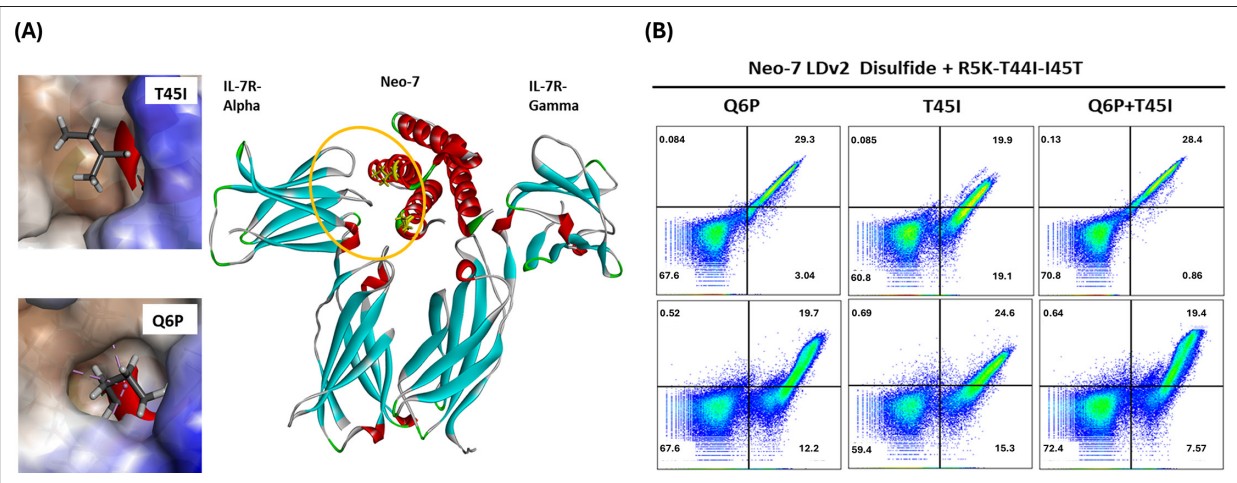

**Figure 4.** Validation of amino acid mutations that confer to the binding affinity of Neo-7 towards IL-7 receptor alpha and gamma. (**A**) Inspection of structural and binding interactions of residue mutations Q6P and T45I on Neo-7 towards the murine IL-7R alpha. (**B**) Yeast display and flow cytometry validation of the binding ability of IL-7/Neo-7 variants toward the IL-7 receptors.

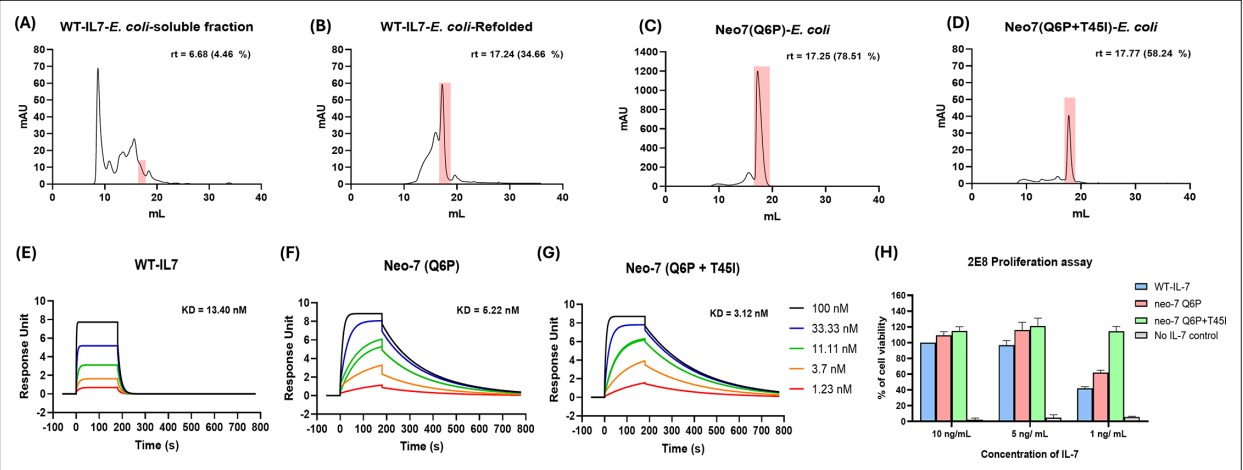

**Figure 5.** Characterization of *E. coli* expressed IL-7 and Neo-7s and the in vitro biological activities. FPLC profile of *E. coli* expressed (**A**) WT-IL7 (**B**) refolded WT-IL7 (**C**) Neo-7-Q6P and (**D**) Neo-7-Q6P-T45I. Percentage of purity is calculated from the SEC-FPLC peak profile via Cytiva Unicorn 7 software after affinity chromatography purification. SPR (Biacore) characterization of the binding kinetics of (**E**) Neo-7-Q6P (**F**) Neo-7-Q6P-T45I and (**G**) WT-IL7 towards murine IL-7R alpha. (**H**) 2E8 proliferation assay to investigate the biological activity of the IL-7/Neo-7s expressed by *E. coli*. Error bars represent standard deviation (n=3).

The online version of this article includes the following figure supplement(s) for figure 5:

**Figure supplement 1.** In silico immunogenicity prediction of (**A**) WT-IL7 (**B**) Neo-7 (**C**) Neo-7 single mutant (**D**) Neo-7 double mutant. (**E**) Sequence comparison of Neo-7 and WT-IL7, helices of the cytokine are colored as blue, green, yellow and red; loops are colored as gray. (**F**) Alphafold predicted structural model of FC-Neo-7.

point of helix 1 that was further stabilized by disulfide stapling and therefore voided the concerns on the helix breaking properties of proline under this design. (3) At position 45, threonine is favored by Rosetta in comparison to the wild type isoleucine residue. Such a decision might arise from the additional consideration of the stability of Neo-7 monomer in aqueous solution as an exposed hydrophobic residue in the aqueous environment is energetically unfavorable. However, after comparing the binding conformations of threonine and isoleucine with IL7Rα, we decided that isoleucine was observed to have better shape complementarity that could occupy the hydrophobic pocket at the Neo-7- IL7Rα interface. Subsequently, we constructed the relevant mutants for Neo-7, sampling the effects of mutations located at the Neo-7-IL7Rα interface towards their binding to IL7Rα. Interestingly, the Q6P mutations showed the most prominent enhancements in the yeast display system, while the T45I mutation showed moderate improvement for IL7Rα. Double mutations of Q6P and T45I further improved the binding affinities of Neo-7 to both IL7Rα and IL2Rγ, suggesting an additive effect of the binding improvement (*Figure 4B*). Two best variants, Neo-7 with Q6P mutation and Neo-7 with dual mutations (Q6*P*+T45 I), were selected for subsequent in vitro bioactivity assay.

## Recombinant Neo-7 shows higher expression and thermostability than WT IL7

After validating Neo-7's ability to bind both IL-7Rα and IL-2Rγ receptors, we aimed to express recombinant Neo-7 in *Escherichia coli*. Wild-type cytokines from the IL-2 family (IL-2, IL-7, IL-15, and IL-21) are often challenging targets for *E. coli* expression due to their complex folding requirements, which

**Table 2.** Yield, purity, and thermostability data of the *E. coli* expressed IL-7/Neo-7s, yield was presented as mg of recombinant protein per 200 mL of *E. coli* culture.

| | Culture volume | Expression mode | Yield after Ni-NTA purification | Yield after FPLC purification | Purity from FPLC (%) | Thermostability (melting temperature) |
|---|---|---|---|---|---|---|
| Neo-7 (Q6P) | 200 mL | Soluble | 3.76 | 1.56 | 78.26 | 71.12 |
| Neo-7 (Q6*P*+T45 I) | 200 mL | Soluble | 1.67 | 0.36 | 58.24 | 70.39 |
| WT-IL7 | 200 mL | Refolding | 7.41 | 0.23 | 23.24 | 56.35 |

necessitate multiple disulfide bonds to stabilize their structures. For instance, WT-IL-7 contains three disulfide bridges, and its soluble expression in *E. coli* yields no detectable properly folded IL-7, as analyzed by FPLC after pre-purification using Ni-NTA affinity chromatography (*Figure 5A–B*). In contrast, Neo-7 was engineered with only one disulfide bridge and designed for more efficient folding and packing compared to WT-IL-7 (*Figure 5C–D*). Indeed, Neo-7 demonstrated superior performance in terms of yield and purity. The Q6P variant of Neo-7 exhibited the highest purity (78.51%) and yield (1.56 mg/200 mL culture), while the double mutation variant (Q6*P*+T45 I) showed lower purity (58.24%) and yield (0.36 mg/200 mL). In comparison, WT-IL-7 was expressed as inclusion bodies, solubilized using 6 M guanidine HCl, and refolded via dialysis, resulting in a final yield of 0.23 mg/200 mL culture with a purity of 34.66% (*Figure 5I*). We also assessed the thermostability of the purified recombinant proteins using the SYPRO assay. As expected, both Neo-7 variants exhibited improved thermostability (71.12°C and 70.39°C for the single and double mutation variants, respectively) compared to WT-IL-7 (56.35°C; *Table 2*). These results confirm that our novel designs confer better protein packing and higher stability than WT-IL-7.

## Recombinant Neo-7 shows enhanced binding affinity to murine IL-7 receptor alpha than WT IL7

We then characterized the binding affinities of Neo-7 variants to mouse IL-7 receptor alpha (mIL-7Rα) in a quantitative manner using surface plasmon resonance (SPR). The dual mutant variant of Neo-7 exhibited the most potent binding to mIL-7Rα, with a dissociation constant (KD) of 3.12 nM, followed by the Q6P variant (KD = 5.22 nM) and WT-IL-7 (KD = 13.4 nM; *Figure 5E–G*). Interestingly, through the SPR curve, we observed different binding kinetics between WT-IL-7 and the Neo-7 variants. WT-IL-7 binds to IL-7Rα through a fast-on-fast-off kinetic model. In contrast, Neo-7 variants displayed a slower on and off rate, with an approximately five times lower on-rates (ka) compared to WT-IL-7 ($1.03 \times 10^6$ and $1.38 \times 10^6$ $M^{-1}s^{-1}$ for Neo-7-Q6P and Neo-7-Q6PT45I, respectively, versus $5.86 \times 10^6$ $M^{-1}s^{-1}$ for WT-IL-7). However, Neo-7 variants also demonstrated significantly slower off-rates (kd) ($6.313 \times 10^{-3}$ and $4.296 \times 10^{-3}$ $s^{-1}$ for Neo-7-Q6P and Neo-7-Q6PT45I, respectively, compared to $7.844 \times 10^{-2}$ $s^{-1}$ for WT-IL-7). This slower dissociation rate compensates for the slower association rate, resulting in more potent overall binding affinities. To assess the biological functions of Neo-7 variants, we then used the IL-7-dependent cell line 2E8 for a cell-based proliferation assay (*Figure 5H*). The ability of the Neo-7 variants in stimulating 2E8 proliferation is correlated with their binding kinetics determined by the SPR assay. At high concentrations (10 ng/mL), the differences in proliferative abilities were insignificant, likely due to saturation of the cytokine. However, at lower concentrations (1 ng/mL), the differences became more pronounced, with Neo-7 variants demonstrating stronger pro-survival signals compared to WT-IL-7. Encouraged by the promising binding affinity data from SPR and 2E8 bioassay, we then submitted the sequence of WT-IL7 and Neo-7s into the Immunomedicine group server for protein immunogenicity prediction. The predicted output showed that WT-IL7 is the most immunogenic with six potential antigenic fragments while the designed Neo-7s had three segments which all overlapped with the WT-IL7 antigenic fragments, indicating no additional immunogenic peptide fragments were introduced throughout the engineering processes of Neo-7 (*Figure 5—figure supplement 1A-D*).

## Fc-fused Neo-7s show superior yield and purity profile in CHO-expression system

Subsequently, we designed a Fc-fused neo-7/WT-IL7 for in vivo anti-tumor assay and expressed the recombinant protein in a CHO-cell based system. Immunomodulatory cytokines are attractive anti-tumor agents due to their immuno-stimulating potential, but the in vivo efficacies are largely limited by their poor pharmacokinetic properties that resulted in rapid clearance from the circulatory system. We therefore fused WT-IL7 and the selected neo-7 variants with the Fc fragments of human IgG1 as a half-life prolonging strategy (*Kwon et al., 2024*; *Foss et al., 2024*; *Martin et al., 2013*). The fused Fc fragments were engineered with LALAPG mutations to attenuate Fc-mediated effector functions to prevent unintended cellular cytotoxicity in the host (*Mausser et al., 2023*). A structural illustration generated by AlphaFold simulation is shown in *Figure 5—figure supplement 1E*. During recombinant protein expression, Fc-fused WT-IL7 exhibited a poorer purity profile (31.84%) compared to the Neo-7 Q6P variant (80.88%) and the double mutation variant (55.15%), indicating that the superior

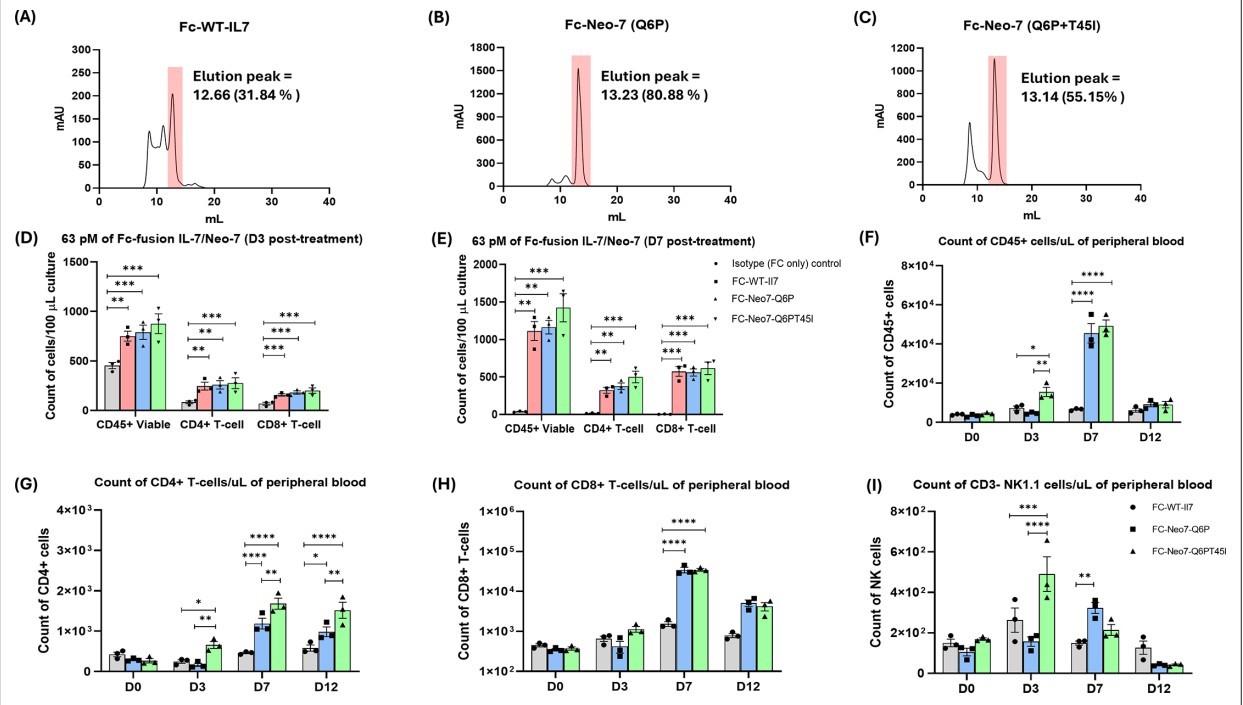

**Figure 6.** Characterization of CHO-S cells expressed FC-fused cytokines and their in vitro and in vivo biological activities. FPLC profile of CHO-S expressed (**A**) WT-IL7 (**B**) Neo-7-Q6P and (**C**) Neo-7-Q6P-T45I. Percentage of purity is calculated from the SEC-FPLC peak profile via Cytiva Unicorn 7 software after affinity chromatography purification. (**D–E**) Murine splenocyte proliferation assays performed at day 3 and day 7 following treatment with Fc-control (gray), Fc-WT-IL7 (red), Fc-Neo-7-Q6P (blue), or Fc-Neo-7-Q6P-T45I (green). In vivo immune stimulatory ability of the Fc-fused cytokines on murine PBMCs at day 0 to day 12 post-treatment. The data are presented as a count of (**F**) total viable CD45+ cells (**G**) viable CD45+ CD3+ CD4+ T cells (**H**) viable CD45+ CD3+ CD8+ T cells (**I**) viable CD45+ CD3- NK1.1+NK cells. Treatment groups are colored as Fc-control (gray), Fc-WT-IL7 (red), Fc-Neo-7-Q6P (blue), and Fc-Neo-7-Q6P-T45I (green). All data were presented as individual data plots with error bars (SEM) (n=3). Statistical differences among groups were determined using one-way ANOVA with Turkey's multiple comparison test. Significance levels are defined as follows *$p < 0.05$; **$p = 0.01$–0.05; ***$p = 0.0001$–0.001; and ****$p < 0.0001$.

production yield and purity of Neo-7 extend beyond prokaryotic systems to mammalian CHO-S cell lines (***Figure 6A–C***). Following FPLC purification, we assessed the bioactivities of the Fc control, Fc-fused IL-7, and Neo-7 variants using a murine splenocyte proliferation assay, with flow cytometry evaluating viability and cell count at Day 3 and Day 7 post-treatment (***Figure 6D–E***). All treatments maintained spleenocyte viability and numbers relative to the Fc control, with a slight but not statistically significant increase in CD45+ cells in the Fc-Neo-7-Q6PT45I group. Our findings demonstrate that Neo-7 variants offer superior yield and purity compared to WT-IL7 in both bacteria (***Table 2***) and mammalian expression systems (***Table 3***).

## Fc-fused Neo-7s shows superior ability to promote T-cell expansions in B6 mice

After validating the biological functions of the Fc-fused cytokines in vitro, we then investigated their in vivo effects on the immune cell composition in mice peripheral blood. Each mouse received a

**Table 3.** Yield and purity data of the CHO-S expressed Fc-fused IL-7/Neo-7s, yields were presented as mg of recombinant protein per 100 mL of CHO cell culture.

| | Culture volume | Expression host | Yield after protein A affinity purification/100 mL culture | Yield after FPLC purification/100 mL culture | Purity from FPLC curve (%) |
|---|---|---|---|---|---|
| Neo-7-Fc | 130 mL | CHO | 15.73 | 10.44 | 80.88 |
| Neo-7 T43I-Fc | 130 mL | CHO | 16.66 | 6.42 | 55.15 |
| WT-IL7-Fc | 130 mL | CHO | 17.22 | 3.37 | 31.84 |

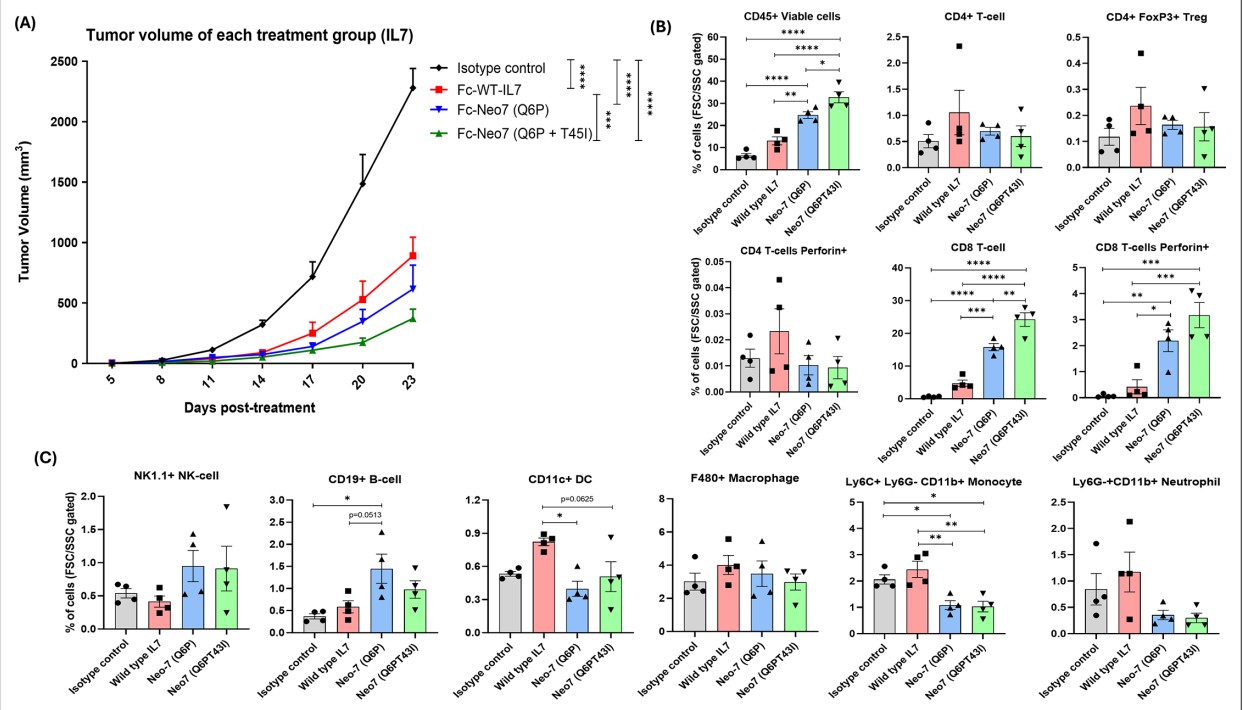

**Figure 7.** In vivo anti-tumor activity of Fc-fused WT-IL7/Neo-7s towards MC38 syngeneic models. (**A**) Tumor growth curve of MC38 tumor after treatment with the Fc-fused cytokines. Proportion of total leukocytes and different immune cell types within the MC38 tumor of mice at day 7 post-treatment. (**B**) Proportion of intratumoral CD4 +and CD8+ T cells (**C**) Proportion of other immune cell types present in the TME at day 7 post-treatment. All data were presented as individual data plots with error bars (SEM). Statistical differences among groups were determined using two-way ANOVA and one-way ANOVA with Turkey's multiple comparison test for tumor growth curve and TILs analysis, respectively. Significance levels are defined as follows *p < 0.05; **p = 0.01–0.05; ***p = 0.0001–0.001; and ****p < 0.0001.

The online version of this article includes the following source data and figure supplement(s) for figure 7:

**Source data 1.** Absolute count of immune cells determined from TILs/mg of the tumor excised.

**Figure supplement 1.** Immune cell profiling of treated mice.

**Figure supplement 2.** Tumor growth, body weight, and treatment schedule.

**Figure supplement 3.** Tumor growth, body weight, and treatment schedule of mice harboring MC38 tumor treated with the combination of Fc-Neo-7 and oxaliplatin.

**Figure supplement 4.** Gating strategy for flow cytometry studies of immune cell profiling conducted in this study.

**Figure supplement 5.** Absolute count of the tumor-infiltrated immune cells.

single intraperitoneal dose of 10 mg/kg Fc-fused IL-7 or Neo-7 variants. Blood samples were collected before treatment (D0) and at 3, 7, and 12 days post-treatment to analyze immune cell composition. We focused on CD45+ cells as an indicator of total immune cells and further examined the subpopulations and numbers of CD3+ CD4+ T cells, CD3+ CD8+ T cells, and CD3- NK1.1+ NK cells, which are the primary effector cell types contributing to the therapeutic effects of cytokine therapies. As shown in *Figure 6F–I*, all treatments exhibited a similar pattern of increased CD45+ immune cells, peaking 7 days post-treatment. Both Fc-fused Neo-7 variants induced significantly higher levels of CD45+ immune cells compared to Fc-WT IL-7 treatment, particularly in the CD8+ T cell subpopulation.

## Fc-fused Neo-7s shows stronger anti-tumor activity in comparison to the WT counterpart

Given the superior immune stimulatory effects of Fc-fused Neo-7s compared to Fc-WT-IL7, we hypothesized that these effects could translate into enhanced anti-tumor activity. To test this, we treated mice bearing subcutaneous MC38 tumors (a syngeneic model for colorectal cancer) with a single dose of 10 mg/kg Fc-fused IL-7 or Neo-7s and monitored tumor growth. All Fc-fused IL-7 treatments showed significant tumor suppression compared to the Fc control (human IgG-1 Fc with LALAPG

mutation), with Fc-fused Neo-7 (Q6PT45I) being the most potent, followed by Fc-fused Neo-7 (Q6P) and Fc-fused WT-IL-7 (*Figure 7A*, *Figure 7—figure supplement 2*).

To explore the possible anti-tumor mechanisms, we conducted a replicate anti-tumor assay and analyzed leukocyte counts and populations in the peripheral blood and tumor of treated mice. We found that the count and proportion of total immune cells and CD3+ CD8+ T cells within the tumor correlated directly with the anti-tumor potency of the treatment groups (*Figure 7C*). In the peripheral blood of tumor-harboring mice, all Fc-cytokine fusions increased the total immune cell count compared to the Fc control, in the order of Fc-Neo-7-Q6P-T45I, Fc-Neo-7-Q6P, and Fc-WT-IL-7 (*Figure 7—figure supplement 1*). This trend was also observed for CD4+ T cells, CD8+ T cells, CD19+ B cells, CD11c+ dendritic cells, and F4/80+ macrophages. Among the immune cells analyzed, CD8+ T cells showed the most significant increase, indicating they might be the primary targets for the immune stimulatory effects of IL-7/Neo-7s (*Figure 7B*). No significant differences were observed in the count of Ly6C+ Ly6g- CD11b+ monocytes in the peripheral blood (*Figure 7—figure supplement 1A*). However, differences were observed between the single (Q6P) and double (Q6P-T45I) mutation variants of Fc-fused Neo-7 in the numbers of CD4+ FoxP3+ Treg and CD3- NK1.1+ NK cells within the peripheral blood. The Q6P variant showed the highest number of Tregs and NK cells among all treatment groups, followed by WT-IL7 and the Q6P-T45I variant. Additionally, only mice treated with Fc-WT-IL7 had significantly higher numbers of Ly6g+ CD11b+ neutrophils compared to other treatments (*Figure 7—figure supplement 1A*). These differences in immune cell regulation might be attributed to variations in binding affinity or kinetics of WT-IL-7 and Neo-7s towards IL-7 receptors on immune cells. Further studies are required to unravel the specific mechanisms underlying this phenomenon.

In the analysis of tumor-infiltrating leukocytes, we found that only CD8+ T cells were significantly enriched in the tumor compared to the Fc control (except for CD19+ B-cells in the Fc-Neo-7-Q6P group), despite a substantial increase of other immune cell types (CD4+ T cells, macrophages, and NK cells) in the peripheral blood (*Figure 7B–C*). This indicates that CD8+ T cells are the primary immune effectors that infiltrate and contribute to the anti-tumor properties of the Fc-Neo-7 fusions. The CD8+ TILs also differed in their effector status, where the perforin-positive CD8+ populations were the highest in the Fc-Neo-7-Q6PT45I group, followed by Fc-Neo-7-Q6P, Fc-WT-IL7, and Fc control. Furthermore, only the Fc-Neo-7 fusions showed a significant reduction of the proportion of the monocytes and neutrophils within the tumor compared to the Fc control, a phenomenon not observed in mice treated with Fc-WT-IL7. The reduction of intra-tumoral monocytes and neutrophils-to-CD8 T cell ratio is a favorable outcome for immunotherapy, as both of these immune cell types have been reported to act as immunosuppressors, inhibiting the cytotoxic activity of CD8+ T cells within the tumor microenvironment (*Figure 7C*; *Sarkar et al., 2023*; *Gallina et al., 2006*; *Ouyang et al., 2024*).

Since IL-7 is also known to promote the generation and maintenance of memory T cells, we also investigated the naive and memory populations of T cells using CD44 and CD62L markers (*Figure 7—figure supplement 1B-C*). Most of the CD4+ T cells circulating in the peripheral blood of the Fc control group were naive T cells (CD62L+ CD44-). However, treatment with Fc-IL-7/Neo-7s increased the number of effector memory (EM) populations (CD62L- CD44+). For CD8+ T cells, treatment with the Fc-cytokine fusion greatly enriched the CD8+ T cell population with a central memory (CM) phenotype (CD62L+ CD44+), followed by naive and effector memory T cells. In the MC38 tumor, most infiltrating T cells were identified as effector memory T cells. The majority of CD4+ tumor-infiltrating T cells exhibited an EM phenotype, with one exception showing a higher number of naive CD4+ T cells in mice treated with Fc-WT-IL7. For tumor-infiltrating CD8+ T cells, nearly all were positive for the memory marker CD44, with the main populations being EM CD8+ T cells (CD62L- CD44+), followed by CM CD8+ T cells (CD62L+ CD44+).

## Fc-Neo-7 (Q6P-T451) greatly enhances expression of genes related to T-cell proliferations and effector functions without up-regulation of exhaustion-related genes

To further investigate the differences between the treatment effects of Fc-Neo-7 (Q6PT43I), Fc-IL-7, and the Fc control, we conducted an RNA-sequencing experiment on the splenic CD8+ T cells of the treated mice. CD8+ T cells were chosen due to their significant responsiveness to the treatments and their abundance in the spleen, ensuring sufficient RNA for sequencing. Principal component analysis

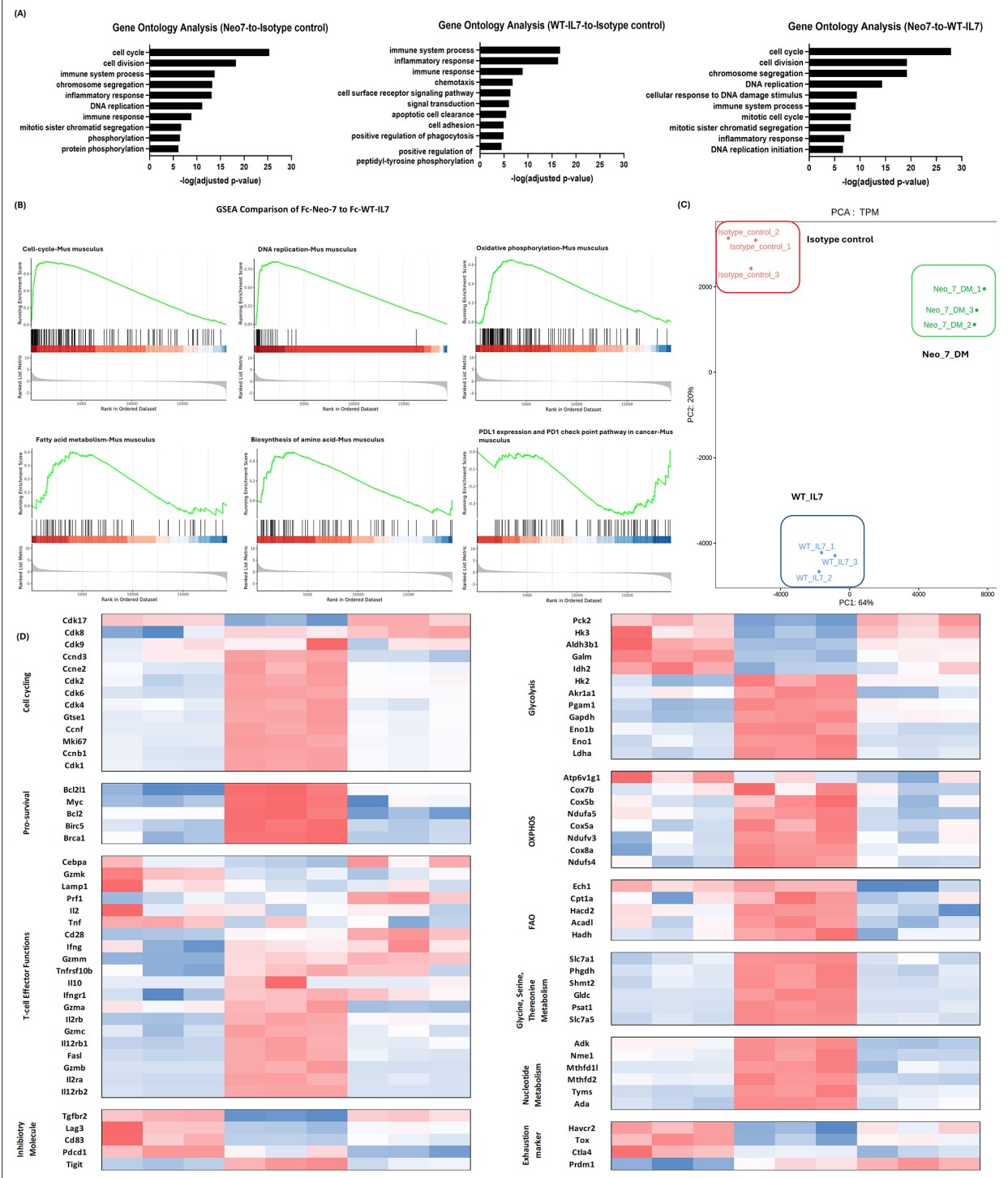

**Figure 8.** RNA sequencing of splenic CD8+T cells isolated from Fc-IL7/Neo-7s-treated mice. (**A**) Gene ontology analysis of the gene expression data from RNA sequencing (n=3; three independent biological donors for each group). (**B**) Gene Set Enrichment Analysis (GSEA) of splenic CD8+T cells treated by Fc-Neo-7 versus Fc-WT-IL7. (**C**) Principal component analysis and (**D**) Gene expression heatmap derived from Z-scores calculated from the RNA sequencing data. The gene expression heatmap is derived from Z-scores calculated from the RNA sequencing data, with expression levels color-coded from high (red) to low (blue).

(PCA) of the RNA-seq data revealed distinct RNA expression profiles for each treatment group, with clear clustering and segregation among the groups (*Figure 8A*). Gene ontology (GO) analysis of the differentially expressed genes showed that both Fc-IL-7 and Fc-Neo-7 upregulated genes related to immune system processes, immune responses, and inflammatory responses relative to the Fc control. Comparing Fc-IL-7 and Fc-Neo-7, we found that genes related to cell proliferation (cell cycle, cell division, and DNA replication) were the most prominently upregulated, corroborating our in vivo mice PBMC proliferation assay data (*Figure 8B*).

Further examination of the gene expression heatmap (*Figure 8C*) showed strong differentiation in the gene expression profile among the two Fc-Neo-7 variants with Fc-WT-IL7. Gene clusters associated with T cell proliferation, fitness, and functionality revealed that Fc-Neo-7 treatment upregulated cell cycle-related genes (Cdk2, Cdk4, Cdk6, Mki67) and pro-survival signals (Bcl-2, Brca1, survivin/BirC5) (*Kim et al., 2020*; *Xue et al., 2010*; *Eum et al., 2024*; *Jacobs et al., 2010*). In addition, a general increase in the expression of genes related to T-cell metabolism was also observed, including those regulating oxidative phosphorylation, fatty acid oxidation, amino acid (glycine, serine threonine pathway), and nucleotide metabolism, indicating an active proliferation and metabolism stage of the CD8 T-cells post-FC-Neo-7 treatments (*Hope and Salmond, 2021*; *Han et al., 2021*; *Shyer et al., 2020*). The serine to glycine one-carbon metabolism pathway was essential to support T cell proliferation by fueling purine biosynthesis and improving mitochondrial respiration efficiency (*Ma et al., 2017*; *Sugiura et al., 2022*). Fc-Neo-7 also primed T cells for enhanced functionality, as evidenced by the upregulation of genes responsible for cytotoxicity (granzymes A, B, and C) and surface receptors for IL-2 (IL2Ra, IL2Rb) and IL-12 (IL12rb), which prepared T cell differentiation into effector types. Notably, while IL-2-induced T cell proliferation often leads to exhaustion and activation-induced cell death (ACID), Fc-Neo-7 treatment resulted in reduced expression of inhibitory molecules (Tgfbr2, Lag3, CD83, PD-1) except TIGIT (*Anderson et al., 2016*; *Gumber and Wang, 2022*). This suggests a potential combination therapy of Fc-Neo-7 with anti-TIGIT to further enhance CD8+T cell effector activity by blocking the immunosuppressive TIGIT pathway. In terms of exhaustion markers, Fc-Neo-7-treated CD8+T cells exhibited marked reductions in the expression of genes such as Tim-3 (Havcr2), Tox, and CTLA4 (*Anderson et al., 2016*; *Gumber and Wang, 2022*). These findings were also confirmed by GSEA analysis, which showed significant enrichment of gene sets related to cell cycle and DNA replication (T cell proliferation), oxidative phosphorylation, and fatty acid metabolism (naive T cell metabolism). In addition, a lower expression of the genes related to the PD1-PDL1 inhibitory axis was also observed (*Figure 8D*).

## Discussion

Cytokines are potent immunomodulatory proteins that serve as attractive therapeutic agents in numerous disease applications. However, clinically, their application is limited due to the need to balance the clinical potency to toxicity and poor pharmacokinetics (*Fu et al., 2023*; *Winer et al., 2022*; *Tang and Harding, 2019*). In this study, we addressed these issues through (1) proper selection of a relatively safe cytokine, IL-7, and (2) redesign of the scaffold of IL-7 via computational protein design to improve the stability and developability of the final molecule, Neo-7. Throughout the design process, we observed that the original structure of the scaffold of WT IL-7 is complex and very inefficient, involving long inter-helices loops that required additional disulfide stapling to ensure proper folding. Since only the helices of IL-7 are required for receptor binding, we chose to truncate all the interconnecting loops and remodel the helices connectivity via Rosetta loop remodeling program. The difference between the design of IL2/15 mimics (Neo 2/15) and this study lies in the design approaches. Neo2/15 was a completely de novo designed protein which required redesigning the mimetics from scratch, while the Neo-7 designed in this study still retains the sequence on the functional-related moiety (receptor binding helices) and thus minimizes the design steps and computational power required (*Silva et al., 2019*). However, it should be noted that our study has a somewhat different objective in that the Neo2/15 is a product from the breakthrough invention of the de novo protein design algorithm from the Baker lab, while the aim of this study is to search for a viable design idea that utilizes the availability of computational protein design (CPD) toolkits such as Alpha-Fold and Rosetta protein design suite for the redesign of a therapeutic-relevant cytokine.

In our approach, we avoided the need for extensive rounds of directed evolution and affinity maturation process. During the design process, we utilize AlphaFold2 to serve as a validation point for our

designed sequence. We observed that the single sequence mode of AlphaFold2 provided a better discriminative power on the designed sequence as the default mode of AlphaFold2 (with reference to the MSA library) provided higher false positive results (bad sequence generating very similar structure to the WT-IL7 helices), possibly due to the highly similar sequences between the WT-IL7 and Neo-7. In addition, we also discovered that the presence of a kink on helix 1 is crucial for the interaction between the IL-7Rα and IL-7/Neo-7. The re-addition of the kink in helix 1 via disulfide stapling and mutation transfer guided by structural visual inspection generated two Neo-7 variants that are more potent in IL-7R binding affinity in comparison to the WT-IL7. Encouragingly, both the Neo-7 variants are also more thermostable compared to the WT-IL-7 and showed a much higher yield and purity. When expressed in *E. coli* (as single entity) and in CHO cells (as Fc-Fusion), indicating that the simpler protein scaffold of Neo-7s is much better handled by the cellular machineries during overexpression. Interestingly, the redesigned scaffold of Neo-7s led to a different binding kinetics towards IL-7Rα in comparison to the WT-IL7. The Neo-7s adopted a slow-on-slow-off model, which we ascribed to the more stable fold as a more flexible (less stable) fold in the WT-IL7 could probably increase the chances to bind to the IL7Rα via an induced-fit mechanism, but the overall flexibilities of the protein will also lead to stronger destabilization of the protein-receptor complex and therefore lead to a faster off-rate (*McElroy et al., 2009*; *Broom et al., 2015*). In contrast, a more stable fold adopted by the Neo-7s would require longer time for the protein-receptor association but also a slower dissociation rate once the complex has been formed (*Celej et al., 2003*; *Majewski et al., 2019*).

Moreover, the comparison of the in vitro and in vivo efficacies of the Fc-Neo-7s and Fc-IL-7 also highlighted the importance of in vivo stabilities during animal studies. In vitro murine spleenocyte proliferation assays showed no significant differences between the samples treated with Fc-Neo-7s and Fc-IL-7, but the in vivo PBMC analysis showed a significant increase in the count of CD8+ T cells (peaked at day 7), possibly due to the better stability of Fc-Neo-7s in the mice circulatory system. Encouragingly, the in vivo immune stimulation activities of the Fc-IL-7/Neo-7s also translated into anti-tumor activities where the efficacy of tumor growth delay was correlated with the binding affinities to IL-7Rα determined in the yeast display and SPR assays. Further analysis of the TILs also revealed a significantly higher infiltration of the CD8+T cells in the tumor, and the higher proportion of Perforin + CD8+ T cells also suggests that the T cells induced by Fc-Neo-7s treatment are functional in terms of cancer cell cytotoxicity. Moreover, the gene expression profile of the murine CD3+ CD8+ spleeno-cytes also revealed that Fc-Neo-7 (Q6PT45I) significantly upregulates genes related to CD8+ T cells proliferation and metabolism without affecting their fitness and functionality, suggesting that Neo-7 (Q6PT45I) is a suitable addition to the current therapeutic space for cancer treatment. To further assess the role of Fc-Neo-7 (Q6PT45I) as an additional arm in anti-cancer therapy, we combined one dose of the Fc-Neo-7 (Q6PT45I) with oxaliplatin, a first-line immunogenic chemotherapy for colon cancer. Indeed, the combination therapy showed a significantly stronger tumor suppressing result in comparison to the untreated control and the monotherapy arms (*Figure 7—figure supplement 3*).

Despite the improved stability, expression yield, and receptor binding affinity achieved by Neo-7, several limitations remain. The present design does not directly mitigate immune toxicity, which is primarily driven by dose-dependent immune activation rather than intrinsic folding inefficiencies, despite the potential toxicity risks being partially mitigated at the design level through the selection of IL-7 as the cytokine scaffold, as IL-7 is generally better tolerated than other γc cytokines such as IL-2 or IL-15 in both preclinical and clinical settings. In addition, while Neo-7 exhibits superior in vivo activity when fused to an Fc domain, its circulating half-life was not extended through scaffold rede-sign alone and still relies on conventional half-life extension strategies. Accordingly, Neo-7 should be viewed as a developability-optimized IL-7 scaffold that is compatible with, rather than a replacement for, pharmacokinetic engineering approaches such as Fc fusion or albumin binding.

In conclusion, we have successfully redesigned IL-7 with a novel folding scaffold termed Neo-7 in this study. Through Rosetta fix backbone modeling and visual inspection, we further identified that the disulfide-induced kink on helix 1 is essential for the binding towards IL-7Rα, and point mutations R5K, Q6P, and T44I could enhance the binding of Neo-7 towards IL-7Rα. The point mutation I54T improved the stability, yield, and purity of Neo-7 but with a slight penalty on the binding strength with the IL7Rα. The designed Neo-7s were also proven to be more potent, stable, and easier to be expressed in both the *E. coli* and CHO systems, highlighting the importance of inherent protein stability in recombinant protein expression. Since the four helical bundle is a common type of protein

fold in the interleukin family (i.e. IL-6, IL-9, IL-21, GMCSF, etc) the concept of the current designs could be further applied into the designs of other types of neo-interleukins to deepen the involvement of computational protein design in the protein therapeutics area and broaden the horizon of interleukin-based therapies in the future.

## Materials and methods

### Cell lines
The following cell lines were used in this study:

- **2E8** (IL-7–dependent murine B cell line; ATCC TIB-239)
- **MC38** (murine colon adenocarcinoma cell line; obtained from Millipore/Sigma-Aldrich, catalog no. SCC172; RRID:CVCL_B288)
- **CHO-S** (Chinese hamster ovary suspension cells; Thermo Fisher Scientific)
- **EBY100** (yeast strain; ATCC)

Identity of the mammalian cell lines (2E8, MC38, and CHO-S) was confirmed by short tandem repeat (STR) profiling, and yeast identity was verified via marker-gene sequencing. Mycoplasma contamination testing was conducted routinely using PCR assays for all mammalian and yeast cell cultures; all results were consistently negative.

### Helix definition and loop remodeling
The PDB structure of human IL-7 with IL-7 receptor alpha (PDB ID: 3DI2) was downloaded and inspected using Pymol visualizer. The helices were defined as follows: helix 1 (residue 8–27), helix 2 (residue 51–66), helix 3 (residue 73–91), and helix 4 (127 -149) adopting the residue numbering of the PDB file (3DI2; *McElroy et al., 2009*). The connecting loops were removed, and each helix was rejoined in a clockwise manner with the orientation of H1>H3>H2>H4. The new loops were remodeled using Rosetta Remodel program according to the developer instructions (*Huang et al., 2011*). The design of the remodeled protein was further optimized via visual inspection and Rosetta fix backbone design according to the developer instructions (here). The amino acid sequence of each protein structure output in the intermediate steps of design was extracted and fed to AlphaFold2 program for protein structure prediction (*Liu et al., 2023*; *Jumper et al., 2021*). The outputs were then aligned to the structural template generated from Rosetta fix backbone design using PyMOL visualizer, and the protein design is accepted when the protein structures generated from both programs aligned well with each other with an RMSD value <1.0 (*Schrödinger and DeLano, 2020*). In this study, despite the human IL-7 being used as a template for Neo-7 design, mouse IL-7 receptors were used for in vitro validations as human IL-7 cross reacts with mouse IL-7 receptors, and such designs ensure the consistency between the in vitro and the in vivo studies.

### Prediction of protein structure using AlphaFold 2 and AlphaFold Multimer
AlphaFold 2 was used for the validation of our protein designs. For monomer prediction, AlphaFold was run using Google Colab with the following settings. First, a non-template and single sequence mode was applied to predict the structure of Neo-7. We hypothesized that since the sequence of Neo-7 is highly similar to the wild type IL-7, the information from PDB structural template and multiple sequence alignment (MSA) might highly favor the generation of the predicted structures that are highly similar to the wild type structure and thus lead to a reduced ability to discriminate the bad structures from the good (https://colab.research.google.com/github/sokrypton/ColabFold/blob/main/Alpha-Fold2.ipynb#scrollTo=G4yBrceuFbf3) (*Mirdita et al., 2022*). On the other hand, predictions using AlphaFold multimer were done in a local workstation with template mode querying through a full database for maximal accuracy of the prediction (https://github.com/google-deepmind/alphafold; *Žídek et al., 2025*; *Liu et al., 2023*; *Jumper et al., 2021*).

### Molecular cloning of neo-7 variants
The amino acid sequences of the selected neo-7 designs were reverse translated into nucleotide sequences using IDT codon optimization tools (https://sg.idtdna.com/CodonOpt) and the transgenes

were synthesized in the form of gBlock followed by cloning into a Aga2-SP based yeast display vector through Gibson assembly. The resulting plasmids were then transformed into *E. coli* (XL10-gold strain) and plated on ampicillin plate (100 µg/mL) for overnight selection. Two colonies were picked and cultured overnight in LB medium supplemented with 100 µg/mL ampicillin for DNA isolation and sequencing.

## Yeast transformation and induction of yeast-displayed proteins

Yeast (strain EBY100) is cultured and maintained in YPD medium consisting of 2% dextrose, 2% peptone, and 1% yeast extract at 30°C with 150 rpm of shaking. A lithium acetate (LiAc) method was applied for yeast transformation. Briefly, yeast was cultured in YPD overnight followed by a 1:50 dilution in fresh YPD medium the next day. The diluted culture was grown to early log phase (OD600=0.2) prior to the transformation. One mL of culture is required for one transformation. The culture was spun at 3000 x *g* for 2 min followed by two rounds of washing, using 1 mL of sterile water and 0.5 mL of LiAc/TE solution (10 mM Tris-HCl, 100 mM LiAc, and 1 mM EDTA). The washed yeast was then pelleted and resuspended with 60 µL of LiAc/TE solution, 25 µL of ssDNA, and 200 ng of the desired plasmid. The mixture was mixed via pipetting followed by addition of 220 µL of PEG solution (40 % w/v PEG3350 dissolved in LiAc/TE solution). The resulting samples were incubated at 30°C for 30 min without shaking, followed by addition of 35 µL of DMSO and 15 min heat shock at 42°C. The yeast suspensions were then pelleted at 3000 x *g* for 1 min, resuspended with 200 µL of sdH$_2$O, and plated on SDCAA plate for drop-out selection. For validation of the display of desired proteins on the transformed yeast, three colonies were selected from the drop-out plate and cultured in SDCAA medium for two passages followed by an 18 hr induction in SGCAA medium (containing 2% galactose as inducing agent).

## Assessing expression level and receptor binding analysis of yeast-displayed neo-7 using flow cytometry

Approximately $1 \times 10^7$ of the induced yeast culture (OD600=1 in 1 mL culture) were pelleted, washed twice with 1X PBS, and resuspended in 100 µL of 1X PBS in a 96-well v-bottom plate. Receptor binding was done by incubation of yeast with a final concentration of 50 nM IL-7 receptor-α (IL7Rα; Sino Biological Cat: 50090-M08H) consisting of a C-terminal polyhistidine tag for 30 min at 4°C on a mini shaker agitated at 600 rpm. The cells were washed and stained with Alexa Fluor 488 anti-HA.11 antibody (Biolegend Cat: 901509; 1:100 dilution) and Alexa Fluor 647 anti-His Tag antibody (Biolegend Cat: 362611; 1:1000 dilution) for an additional 15 min at 4°C to determine the expression level of the displayed protein and their binding affinity towards IL-7 receptor-α. To assess the binding of the common IL-2 receptor γ (IL2Rγ) to neo-7 and IL7Rα complexes, the yeasts were first incubated with 50 nM of IL7Rα, washed, and incubated with 100 nM of human-Fc tagged IL-2 receptor-γ (IL2Rγ; Sino Biological Cat: 50087-M03H) at 4°C for an hour. The yeast was then washed twice with 1X PBS and stained with Alexa Fluor 647 anti-HA.11 antibody (Biolegend Cat: 682404; 1:200 dilution) and PE anti-human IgG-Fc antibody (Abcam cat: ab131612; 1:10 dilution) for an additional 15 min to determine the expression level of the displayed protein and their binding affinity towards IL-2 receptor-γ. The yeasts were then spun down, washed twice with 1X PBS, and assessed by flow cytometry. Flow cytometry data were analyzed using FlowJo software. Yeast cells were first gated based on forward and side scatter to exclude debris (FSC-A vs. SSC-A), followed by singlet gating (FSC-A vs. FSC-H). Surface expression and binding were assessed using aforementioned fluorescent markers, and double-positive populations were quantified. The full gating strategy is shown in *Figure 3—figure supplement 1*.

## Expression and purification of recombinant Neo-7 in *E. coli* (BL21)

The desired neo-7 gene designs were PCR amplified to generate amplicons with at least 20 bp overlapping with the linearized bacteria expression vector. The incorporated transgenes were flanked by a ST-II signal tag and a hexa-histidine tag at their 5' and 3' ends to target the recombinant protein into the periplasmic region for proper disulfide formation and to aid purification, respectively. The recombinant plasmid was transformed into *E. coli* BL21 competent cells and grew in LB-kanamycin (50 µg/mL) medium for selection of transformant. On the day of protein expression, overnight culture of the bacteria was diluted (1:200) in fresh 2X-YT medium supplemented with 25 µg/mL kanamycin

and grown into log phase (OD600=0.4–0.6) at 37°C. The cultures were then transferred into a chilled incubator (18°C) and incubated for an additional 48 hr for protein expression (no IPTG is required for expression). After incubation, the cultures were pelleted at 8000 x $g$ for 20 min, supernatant was discarded, and the pellets were stored in –80°C for overnight. The recombinant protein was purified using nickel NTA affinity chromatography. The frozen pellets were thawed on ice for an hour, followed by cell lysis using B-PER Bacterial Protein Extraction Reagent according to manufacturer protocol. The supernatant was collected and incubated with the Ni-NTA resins (125 µL of resin for 100 mL of bacteria culture) at 4°C for an hour. The Ni-NTA column was washed with wash buffer until no A280 signal was observed from the eluate. The recombinant proteins were then eluted with 5 mL of elution buffer. The eluate was then spun in an Amicon Ultra-15 centrifugal filter (3 kDa) to exchange the elution buffer with 1X PBS (pH 7.4) and to concentrate the protein into a suitable volume size (600 µL) for FPLC purification using Superdex 200 Increase 10/300 GL size exclusion chromatography column. On the other hand, a similar procedure was used for expression of WT-IL7, except that WT-IL7 was expressed as an inclusion body and the inclusion body was solubilized using 6 M guanidine-HCl, purified using Ni-NTA affinity chromatography, and refolded in refolding buffer (100 mM Tris-HCl, 1 mM EDTA, 250 mM L-arginine, 1 mM reduced glutathione, 0.5 mM oxidized glutathione, 10 % v/v glycerol, pH 8.0) overnight at 4°C. The refolded protein was dialyzed 1 round against 100 mM Tris-HCl (pH 10) and another three rounds against 1X PBS (pH 7.4) to remove the remaining additives. The refolded protein was concentrated and purified using FPLC for subsequent in vitro assays.

## In vitro validation of neo-7 bioactivity

### 2E8 proliferation assay

2E8 cells (ATCC TIB-239), an IL-7-dependent murine B-cell cell line, were cultured in Iscove's modified Dulbecco's medium (IMDM; Gibco Cat: 12440053) supplemented with 0.05 mM 2-mercaptoethanol 2 ng/mL mouse IL7 (Sino Biological) and 20% FBS. Proliferation assay was conducted using CCK-8 assay kit. Briefly, $1 \times 10^5$ 2E8 cells were incubated in 100 µL of medium containing 1 ng/mL and 10 ng/mL IL-7 in a flat bottom 96-well plate for 3 days in a humidified $CO_2$ incubator at 37°C. 10 µL of the CCK-8 reagent was added into each well followed by an additional 3 hr incubation at 37°C. The resulting A450 signal was then read using Tecan infinite microplate readers and all data were analyzed using GraphPad Prism 8.0.

### Murine spleenocyte proliferation assay

Spleens from C57BL/6 J mice were harvested and ground against a 40 µm strainer for spleenocyte isolation. The isolates were pelleted, followed by lysis of RBC and cell counting. The spleenocytes were cultured in RPMI medium supplemented with 10% heat-inactivated FBS, 0.05 mM β-mercaptoethanol, 10 mM HEPES, 2 mM glutamine, 0.1 mM nonessential amino acids, 1 mM sodium pyruvate, and 1X insulin-transferrin-selenium (ITS). For the proliferation assay, $2 \times 10^6$ spleenocytes were incubated in a 12-well plate with 1 mL of the culture medium. All spleenocytes were treated with 1 µg/mL of anti-CD3 and anti-CD28 antibodies on the first day for T-cell stimulation, supplemented with either 2 ng/mL or 20 ng/mL of WT-IL7 or neo-7 variants. The resulting spleenocytes were counted using an automated cell counter and split at a 1:5 ratio every three days into fresh medium containing freshly prepared IL-7.

### Surface Plasmon Resonance (SPR) analysis of neo-7 binding affinity to murine IL7Rα

SPR was conducted to determine the binding affinity of recombinant neo-7 variants to murine IL7Rα. Neo-7 recombinant proteins were produced via *E. coli* as mentioned in the previous section. Recombinant murine IL7Rα was diluted to 20 µg/mL in 10 mM sodium acetate (pH 4.5) and immobilized to a CM5 CHIP (Cytivia) via amine coupling. The maximum response unit (Rmax) was controlled to be below 30 to reduce mass transport limitations according to the manufacturer protocol. A threefold serial dilution of WT-IL7 and two neo-7 variants was performed in HBS-EP+ buffer and flowed past (50 µL/min) the immobilized chip with an association and dissociation time of 180 s and 600 s respectively. The chip was regenerated by 30 s exposure to 3 M $MgCl_2$ with a flow rate of 10 µL/min. The binding affinity (KD) and dissociation constants (Kd) were calculated using Biacore T200 evaluation software and the resulting binding curves were regenerated using GraphPad Prism 8.0.

## In vivo validation of neo-7 bioactivity

### Design and expression of recombinant Neo-7-Fc fusion

Following the validation of the bioactivity of Neo-7, we proceeded to construct a Neo-7-Fc fusion protein by fusing Neo-7 variants to the C-terminus of a human Fc fragment (with LALAPG mutation) using a 20-amino acid glycine-serine linker $(GGGGS)_4$. The gene construct was then cloned into the pCDNA 3.4 vector and transfected into Chinese hamster ovary (CHO) cells for recombinant protein expression. The Expifectamine CHO transfection kit was employed for transfection, following the manufacturer's protocol. In brief, 100 μg of plasmid DNA was transfected into 100 mL of CHO cells pre-grown to a density of $6 \times 10^6$ cells/mL. CHO-enhancer and feed were added at 16 hr post-transfection. The culture was allowed to grow for an additional 10 days at 8% $CO_2$ with 150 rpm shaking. Subsequently, the recombinant protein was purified using protein-G affinity chromatography, concentrated with an Amicon tube protein concentrator, and further purified to achieve at least 95% purity using FPLC chromatography. The resulting protein was eluted with 1X PBS (final concentration = 1 mg/mL), flash-frozen using liquid nitrogen, and stored at –80°C for future use.

### In vivo validations of Neo-7 biological functions

All mice used in this study were wild type C57BL/6 (B6) strain and were procured from the National Laboratory Animal Center (NLAC), NAR Labs, Taiwan. The mice were housed at the animal facilities of the Institute of Biomedical Sciences, *Academia Sinica*, with maintained environmental conditions, including proper temperature (19–23°C) and humidity (50–60%), under a 12 hr light-dark cycle. All animal experiments were conducted at the animal house of the Institute of Biomedical Sciences, *Academia Sinica*, in compliance with scientific and ethical guidelines, as approved by the institution's Animal Care and Usage Committee.

### Peripheral blood immune cells analysis in murine model

B6 mice received intraperitoneal injections of different variants of Neo-7-Fc fusions to assess the in vivo biological effects of the designed proteins on the immune composition in mice peripheral blood. Each mouse was administered 10 mg/kg of the respective treatments (Fc-only; Fc-WT-IL7; Fc-Neo-7-Q6P; Fc-Neo-7-Q6PT45I). Peripheral blood samples were collected via submandibular blood collection before treatment, as well as 3 days, 7 days, and 12 days post-treatment. The collected blood was preserved in an EDTA tube (BD microcontainer) and subjected to immediate red blood cell (RBC) lysis, followed by flow cytometry analysis to assess leukocyte populations.

### In vivo anti-tumor activity of Neo-7-Fc fusion protein in MC38 syngeneic models

To induce tumor in mice, 1 X 105 of MC38/CT26 cells were lifted with versene, suspended in 1X PBS, and subcutaneously injected into the right flank of each mouse. Treatment was initiated at day 5 post-tumor inoculation for tumor growth analysis and day 10 for combination treatment and TILs analysis. Tumor volume (mm3) was calculated using the following equation: [(tumor length + tumor weight)/2]3 x 0.52. Mice with a tumor that was larger than 1500 mm3 were considered dead for ethical reasons.

### Tumor-infiltrating leukocytes (TILs) analysis

The tumors from mice in different treatment groups were excised, weighted, and processed to obtain tumor cell suspensions. The tumors were cut into small pieces, immersed in TILs buffer (composed of 670 μg/mL collagenase IV and 2 μg/mL DNase I dissolved in MC38 culture medium) and processed into single-cell homogenates using the gentleMACS Octo Dissociator (Miltenyi Biotec Inc). The resulting cell suspensions were passed through a 40 μm cell strainer, treated with 1X RBC lysis buffer, and resuspended in FACS buffer (5% FBS in PBS, v/v, pH 7.2). All samples were blocked with CD16/32 Fc blocker (1: 50 dilution) prior to immunostaining. For surface staining, the TILs were stained with eFluor780 eBioscience-Fixable Viability Dye, Pacific Blue-CD45 antibody, PerCP/Cyanine5.5-CD45 antibody, APC-CD4 antibody, AlexaFluor-CD8a antibody, Alexa Fluor647-Ly6G antibody, FITC-CD11B antibody, Pacific Blue-CD11B antibody, BrilliantViolet421-F4/80 antibody, and FITC-MHCII antibody. For intracellular staining, all samples were fixed and permeabilized with 4% paraformaldehyde (PFA) and 0.1% saponin, respectively, after staining of the surface markers. The samples were then stained

with PE-IFN-γ antibody and Brilliant Violet 785-TNF-α antibody. The stained samples were analyzed using Attune NTX flow cytometer, and the data were processed using FlowJo V10 software. The full gating strategy is shown in *Figure 7—figure supplement 4* and the absolute count of the tumor-infiltrated immune cells is provided in *Figure 7—figure supplement 5*.

## RNA sequencing of splenic CD8+ T-cells

B6 mice were intraperitoneally treated with 10 mg/kg of the Fc-fused cytokines (Fc-only Fc control; Fc-WT-IL-7 and Fc-Neo-7-Q6PT45I). At day 7 post-treatment, mice were sacrificed and splenic CD8+ T cells were isolated using EasySep Mouse CD8+ T cell isolation kit according to the manufacturer protocol. The isolated CD8+ T cells were then lysed using TRIzol reagent and sent to Genomics Inc (Taiwan) for RNA isolation and RNA sequencing.

## Acknowledgements

The work was supported by the National Science and Technology Council of the Republic of China (NSTC-110–2113 M-001-064-MY3) and Academia Sinica, Taiwan (AS-CDA-108-L07).

## Additional information

### Competing interests

See-Khai Lim, Che-Ming Jack Hu: is listed as an inventor on a patent application related to the present study (Academia Sinica Docket # 12A-1131122). The other authors declare that no competing interests exist.

### Funding

| Funder | Grant reference number | Author |
|---|---|---|
| National Science and Technology Council | 110-2113-M-001-064-MY3 | Kurt Yun Mou |
| Academia Sinica | AS-CDA-108-L07 | Kurt Yun Mou |

The funders had no role in study design, data collection and interpretation, or the decision to submit the work for publication.

### Author contributions

See-Khai Lim, Conceptualization, Data curation, Formal analysis, Validation, Investigation, Methodology, Writing – original draft, Writing – review and editing; Wen-Ching Lin, Data curation, Validation, Investigation, Methodology; Yi-Chung Pan, Sin-Wei Huang, Data curation, Formal analysis; Yao-An Yu, Cheng-Hung Chang, Data curation, Formal analysis, Investigation; Che-Ming Jack Hu, Resources, Supervision, Validation, Project administration, Writing – review and editing; Chung-Yuan Mou, Supervision, Project administration, Writing – review and editing; Kurt Yun Mou, Conceptualization, Resources, Data curation, Formal analysis, Funding acquisition, Investigation, Methodology, Writing – original draft, Project administration

### Author ORCIDs

See-Khai Lim ⓘD https://orcid.org/0000-0001-6031-849X
Sin-Wei Huang ⓘD https://orcid.org/0009-0004-1593-6511
Che-Ming Jack Hu ⓘD https://orcid.org/0000-0002-0988-7029
Chung-Yuan Mou ⓘD https://orcid.org/0000-0001-7060-9899

### Ethics

All animal experiments were conducted in accordance with the guidelines for the Care and Use of Laboratory Animals and were approved by the Institutional Animal Care and Use Committee (IACUC) of the Institute of Biomedical Sciences, Academia Sinica, Taiwan. The approved protocol number was (#15-10-868). All mice were housed under specific pathogen-free conditions with controlled temperature, humidity, and a 12-hour light/dark cycle. All procedures, including tumor implantation

and administration of test compounds, were performed under approved protocols, and every effort was made to minimize animal suffering.

Reviewer #1 (Public review): https://doi.org/10.7554/eLife.107671.3.sa1
Author response https://doi.org/10.7554/eLife.107671.3.sa2

## Additional files

### Supplementary files
MDAR checklist

### Data availability
All data generated or analysed during this study are included in the manuscript and supporting files. Source data can be accessed at Dryad (https://doi.org/10.5061/dryad.9zw3r22v5).

The following dataset was generated:

| Author(s) | Year | Dataset title | Dataset URL | Database and Identifier |
| --- | --- | --- | --- | --- |
| Lim SK, Mou CY, Hu CM | 2025 | Data from: Targeted computational design of an interleukin-7 superkine with enhanced folding efficiency and immunotherapeutic efficacy | https://doi.org/10.5061/dryad.9zw3r22v5 | Dryad Digital Repository, 10.5061/dryad.9zw3r22v5 |

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
