## [Editor Report · eLife Assessment]

This **important** study presents the rational redesign and engineering of interleukin-7. The data from the integrated approach of using computational, biophysical, and cellular experiments are **convincing**. This paper is broadly relevant to those studying immunomodulation using biologics.

---

## [Referee Report · Reviewer #1 (Public review)]

Summary:

This manuscript describes the use of computational tools to design a mimetic of the interleukin-7 (IL-7) cytokine with superior stability and receptor binding activity compared to the naturally occurring molecule. The authors focused their engineering efforts on the loop regions to preserve receptor interfaces while remediating structural irregularities that destabilize the protein. They demonstrated the enhanced thermostability, production yield, and bioactivity of the resulting molecule through biophysical and functional studies. Overall, the manuscript is well written, novel, and of high interest to the fields of molecular engineering, immunology, biophysics, and protein therapeutic design. The experimental methodologies used are convincing; however, the article would benefit from more quantitative comparisons of bioactivity through titrations.

Comments on revisions:

All comments have been sufficiently addressed, with the exception of comment 24 from Reviewer 1. The authors need to modify the manuscript abstract, introduction, and/or discussion to clarify which limitations of IL-7 were addressed by their molecule and to note the limitations of their approach in terms of mitigating toxicity or enhancing half-life.

---

## [Author Response]

The following is the authors’ response to the original reviews.

**Public Reviews:**

**Reviewer #1 (Public review):**
Summary:This manuscript describes the use of computational tools to design a mimetic of the interleukin-7 (IL-7) cytokine with superior stability and receptor binding activity compared to the naturally occurring molecule. The authors focused their engineering efforts on the loop regions to preserve receptor interfaces while remediating structural irregularities that destabilize the protein. They demonstrated the enhanced thermostability, production yield, and bioactivity of the resulting molecule through biophysical and functional studies. Overall, the manuscript is well written, novel, and of high interest to the fields of molecular engineering, immunology, biophysics, and protein therapeutic design. The experimental methodologies used are convincing; however, the article would benefit from more quantitative comparisons of bioactivity through titrations.
**Reviewer #2 (Public review):**
Summary:This manuscript presents the computational design and experimental validation of Neo-7, an engineered variant of interleukin-7 (IL-7) with improved folding efficiency, expression yield, and therapeutic activity. The authors employed a rational protein design approach using Rosetta loop remodeling to reconnect IL-7's functional helices through shorter, more efficient loops, resulting in a protein with superior stability and binding affinity compared to wild-type IL-7. The work demonstrates promising translational potential for cancer immunotherapy applications.Strengths:(1) The integration of Rosetta loop remodeling with AlphaFold validation represents an established computational pipeline for rational protein design. The iterative refinement process, using both single-sequence and multimer AlphaFold predictions, is methodologically sound.(2) The authors provide thorough characterization across multiple platforms (yeast display, bacterial expression, mammalian cell expression) and assays (binding kinetics, thermostability, bioactivity), strengthening the robustness of their findings.(3) The identification of the critical helix 1 kink stabilized by disulfide bonding and its recreation through G4C/L96C mutations demonstrates deep structural understanding and successful problem-solving.(4) The MC38 tumor model results show clear therapeutic advantages of Neo-7 variants, with compelling immune profiling data supporting CD8+ T cell-mediated anti-tumor mechanisms.(5) The transcriptomic profiling provides valuable mechanistic insights into T cell activation states and suggests reduced exhaustion markers, which are clinically relevant.Weaknesses:(1) While computational predictions are extensive, the manuscript lacks experimental structural validation of the designed Neo-7 variants. The term "Structural Validation" should not be used in the header.

We thank the reviewer for this constructive comment. To better reflect the work conducted, we have revised the section title from “Structural Validation of Neo-7 in AlphaFold single sequence mode” to “Structural Modeling of Neo-7 in AlphaFold single sequence mode.” This change clarifies that our study employed in silico modeling approaches rather than experimental structural validation.

We thank the reviewer for this insightful comment. We speculate that the slower off-rate observed for Neo-7 variants is primarily attributable to their enhanced structural stability, which promotes the formation of a more stable cytokine–receptor complex. This is consistent with prior observations in other engineered cytokines, such as IL-2 mimetics (Neo-2/15).

In terms of biological consequences, we believe the slower off-rate is unlikely to result in signaling bias or qualitatively distinct pathways for several reasons:

IL-7’s mechanism of action is inherently regulated to prevent over-signaling. T cells downregulate IL7R-α expression upon IL-7 stimulation, ensuring a built-in negative feedback mechanism.

IL-7 signaling is dominated by STAT5 activation, without the signaling plasticity observed in cytokines like IL-21 or IL-22, which can bias toward STAT1/3 and drive divergent functional outcomes.

Our RNA-seq data support this interpretation, as Neo-7–treated CD8⁺ T cells exhibited transcriptional profiles highly similar to those induced by WT-IL-7, with the difference being an enhanced magnitude of response rather than novel pathway engagement.

Taken together, we infer that the slower off-rate of Neo-7 enhances the potency and durability of IL-7 signaling without altering its downstream specificity, thereby strengthening the magnitude of immune responses while maintaining the canonical STAT5-driven biology of IL-7.

(3) While computational immunogenicity prediction is provided, these methods are very limited.

We fully agree with the reviewer that current in silico immunogenicity prediction tools are limited and cannot be considered definitive. Indeed, to date, none of these algorithms has demonstrated a strong correlation with clinical immunogenicity outcomes of biologics. For example, the presence of anti-drug antibodies (ADA) in murine or non-human primate models often does not translate into ADA induction in human clinical trials. This disconnect underscores the inherent challenges of predicting immunogenicity based solely on computational or preclinical models.

Our strategy to mitigate potential immunogenicity was therefore not to rely exclusively on prediction software, but instead to apply a conservative design principle: preserving the vast majority of the parental IL-7 sequence while introducing only the minimal number of amino acid substitutions required to achieve our engineering objectives. By maintaining sequence continuity with the native cytokine, we aim to minimize the risk of introducing novel epitopes while improving stability and developability. We acknowledge that definitive immunogenicity assessment can only be addressed in future clinical studies.

**Recommendations for the authors:**

**Reviewer #1 (Recommendations for the authors):**
Specific Points:(1) The authors should describe the molecular composition of CYT-107.

We thank the reviewer for this suggestion and have added clarification regarding the molecular composition of CYT-107. CYT-107 is a recombinant form of wild-type human interleukin-7 (IL-7) expressed in eukaryotic cells, which introduces N-linked glycosylation modifications to the protein. As a glycosylated recombinant IL-7, CYT-107 more closely mimics the natural human cytokine compared to bacterial expression systems that produce non-glycosylated IL-7. This feature contributes to its stability and bioavailability in clinical applications.

(Reference: U.S. National Center for Advancing Translational Sciences, GSRS record for IL-7, https://gsrs.ncats.nih.gov/ginas/app/ui/substances/46bd8013-1e2d-4b6e-afcf-340f447e8710)

(2) The authors should indicate the receptor layout for IL-7 in the introduction and indicate available structural data. Also, in line 93, the authors should indicate that IL-7Ra is one subunit of the heterodimeric receptor complex.

We thank the reviewer for this insightful suggestion. However, due to page limitations, we have chosen to orient the introduction around the design rationale, computational workflow, and biological functionality of IL-7. To address the reviewer’s point while maintaining brevity, we have now included a concise description of the IL-7 receptor layout and its available structural data in the main text. Specifically, in line 93 we revised the sentence to read:“We began by examining the crystal structure of IL-7 bound to its receptor, IL7R-α (interleukin-7 receptor alpha; PDB ID: 3DI2), which recruits IL-2Rγ to form a heterodimeric receptor complex essential for downstream signaling.”

(3) The abbreviation IL-7Ra should be defined at first use.

We thank the reviewer for the comment. The abbreviation has now been defined at its first appearance in the manuscript. Specifically, at Line 93 we revised the sentence as follows:

“We began by examining the crystal structure of IL-7 bound to its receptor, IL7R-α (interleukin-7 receptor alpha; PDB ID: 3DI2), which recruits IL-2Rγ to form a heterodimeric receptor complex essential for downstream signaling..”

(4) The authors need to clarify whether the human or murine IL-7Ra is being used in each experiment mentioned in the results text.

We thank the reviewer for this important point. We have now specified in the main text and corresponding subsection titles whether human or murine IL-7Rα was used in each experiment.

(5) The authors sometimes use a dash in IL7Ra and IL2Rg and sometimes do not. This should be standardized.

We appreciate the reviewer’s observation. We have standardized the terminology throughout the manuscript to “IL7Rα” and “IL2Rγ” to maintain consistency.

(6) In Figure 3E, the authors left out the v in "Neo7-LDv1".

We have corrected the omission of “v” and updated the label to read Neo7-LDv1.

(7) In Figure 3E, the authors must indicate in the bottom row that they are visualizing sequential binding to IL-2Rg following incubation with IL-7Ra. This should be stated in the results text and the figure caption as well.

We have revised the results text and figure caption to clearly state that the bottom row illustrates sequential binding to IL-2Rγ following incubation with IL-7Rα.

“for detection of IL-2Rγ binding, yeast cells were first incubated with recombinant IL-7Rα, washed, and subsequently incubated with IL-2Rγ”

(8) In Figure 3E, "IL-7Rg" should be corrected to "IL-2Rg".

We have corrected “IL-7Rγ” to “IL-2Rγ” in Figure 3E for accuracy and consistency.

(9) In line 140, the authors claim that Neo7-LDv1 is partially folded based on the binding to the heterodimeric receptor complex. However, the data are insufficient to support this conclusion.

We understand the concern of the reviewer and we decided to rephrase the sentence for better understanding: “A degree of binding to IL2Rγ was detected, possibly reflecting partial folding of the displayed protein in the yeast display platform.” While we do not claim the protein to be fully or uniformly folded, this deduction is supported by the yeast display data and further corroborated by AlphaFold structural predictions.

(10) In lines 185-186, the authors claim that the binding affinity for IL-2Rg is improved, but this is not shown in Figure 3, which looks only at a single concentration and shows comparable binding between WT-IL7 and Neo7-LDv2.

We thank the reviewer for this valuable observation. Our original wording was ambiguous and may have implied a direct comparison with WT-IL7, which was not intended. The sentence was meant to highlight that within the Neo-7 variant series, Neo7-LDv2 displayed stronger binding to both IL-7Rα and IL-2Rγ compared to other Neo-7 variants. To avoid misinterpretation, we have revised the text as follows:

“Importantly, the enhanced binding affinity towards IL7Rα also led to improved binding towards the common IL2Rγ., relative to other variants in the Neo-7 series.”

(11) Lines 202-203 appear to be an error.

We thank the reviewer for pointing this out. The lines in question were indeed an error and have now been removed from the manuscript.

(12) In yeast display validation, negative controls showing binding to the fluorescent antibody only and an irrelevant control protein should be shown for all constructs in order to evaluate nonspecific interactions.

We agree with the reviewer that appropriate negative controls are important to validate specificity. To address this, we will include yeast display data with negative controls—native yeast (EBY100) stained with the corresponding fluorescent antibody in the Supplementary Information. This addition will provide clearer validation of binding specificity and reduce concerns regarding nonspecific interactions.

(13) For yeast display studies, titrations rather than single concentrations should be used to compare constructs (Figures 3 and 4). The claim that any of the constructs has a higher affinity than any other construct must be supported by performing titrations.

We thank the reviewer for this comment. We respectfully note that yeast display titrations provide relative rather than absolute estimates of binding affinity. In our study, constructs were compared under identical antigen concentrations, where the observed fluorescence intensity reflected their relative binding strength. These yeast display results served as an initial screening strategy, which we subsequently validated using surface plasmon resonance (SPR). SPR provided quantitative binding parameters and confirmed the binding differences observed in yeast display. Thus, while yeast titrations were not performed, the combination of side-by-side yeast display comparisons and orthogonal validation by SPR supports our affinity claims with both qualitative and quantitative evidence.

(14) The acronym SPR needs to be defined, and the authors should mention that this technique was used for quantitative binding studies in line 259.

We thank the reviewer for this suggestion. The acronym has now been defined in the main text at its first use, and we have clarified its role in the study. The revised text reads:

“We then characterized the binding affinities of Neo-7 variants to mouse IL-7 receptor alpha (mIL-7Rα) in a quantitative manner using surface plasmon resonance (SPR).”

(15) A titration of 2E8 cell proliferation versus concentration should be presented for IL-7 versus Neo-7 variants to directly compare EC50 values and make claims regarding potency in Figure 5H. Also, the authors should clarify whether a proliferation or viability assay was performed.

We thank the reviewer for the helpful comment regarding the use of EC₅₀ values when discussing potency. In response, we have revised the manuscript to avoid overinterpreting the data. Specifically, we replaced the term potency with ability to stimulate, as the 2E8 cell assay was designed to validate whether receptor binding by IL-7 and Neo-7 variants translates into biological function—namely, supporting immune cell viability and proliferation under limiting cytokine conditions. The assay was not optimized to determine formal EC₅₀ values, but rather to demonstrate functional activity consistent with IL-7 receptor engagement.

We have also clarified in the text that the experiment was a proliferation assay, with cell viability assessed as part of the readout. This revision better reflects the scope of the assay while aligning our claims with the data presented.

(16) Isotype control is not an appropriate name for the Fc-Only construct. This should be denoted as Fc Only.

We thank the reviewer for this comment. We have revised the terminology throughout the manuscript, changing isotype control to Fc control.

(17) A titration of mouse splenocyte proliferation versus concentration should be presented for IL-7 versus Neo-7 variants to directly compare EC50 values and make claims regarding potency in Figure 6.

We thank the reviewer for this insightful suggestion regarding EC₅₀ analysis. In this study, the splenocyte proliferation assay was designed as a preliminary in vitro screen to confirm the biological activity of Neo-7 variants relative to wild-type IL-7 prior to in vivo testing. The assay was not optimized for quantitative potency determination, but rather to provide an initial functional validation of the constructs. We have therefore revised the manuscript wording to avoid overinterpreting the data and refrained from making claims regarding EC₅₀-based potency. Instead, we emphasize that the in vivo tumor model provides a more physiologically relevant and rigorous platform for assessing cytokine functionality, including proliferation and immunomodulation.

(18) The legends in Figure 6 should indicate the colors used for each construct.

We thank the reviewer for pointing this out. We have revised the legend for Figure 6 to include the color codes corresponding to each construct.

(19) Metabolism should be singular in lines 433 and 435.

We have corrected the wording so that “metabolism” is consistently used in the singular form.

(20) In Figure 8D, "cycling" should be changed to "cycle".

The word “cycling” has been corrected to “cycle” in Figure 8D.

(21) The treatments need to be indicated in Figure 8D. Also, a color scale is needed.

We agree with the reviewer, and a color scale description has now been included in the Figure legend to aid interpretation. “The gene expression heatmap is derived from Z-scores calculated from the RNA sequencing data, with expression levels color-coded from high (red) to low (blue). ”

(22) More comparisons between RNASeq data for Fc-WTIL7 versus Fc-Neo7 (Figure 8) should be presented in the results section.

We thank the reviewer for this suggestion. Due to space limitations in the main manuscript, we are unable to include an expanded description of all RNA-Seq comparisons. However, we will provide a more detailed analysis of Fc-WT-IL7 versus Fc-Neo7 in the supplementary section, including expanded differential gene expression comparisons and pathway enrichment analyses. This will allow readers to fully appreciate the differences while maintaining focus in the main text.

(23) The strikethrough in line 464 needs to be corrected.

We have corrected the strikethrough error in line 464.

(24) It is unclear how stabilizing IL-7 improves its toxicity or half-life. The authors should indicate more clearly which limitations of IL-7 were addressed by their molecule in the abstract, introduction, and discussion.

Native IL-7 demonstrates an excellent safety profile but faces two major challenges in clinical application: (1) short plasma half-life and (2) suboptimal developability due to poor stability. The short half-life is typically addressed through Fc-fusion strategies, which extend systemic exposure via FcRn recycling. However, wild-type IL-7 exhibits a strong aggregation tendency when fused to Fc, rendering the fusion protein poorly developable. By redesigning IL-7 into the more stable Neo-7 format, we substantially improved the folding efficiency and purity of the Fc-fusion protein after affinity purification, thereby enabling its advancement as a recombinant biologic candidate.

We do not intend to claim that increased stability directly reduces in vivo toxicity. The favorable safety profile of IL-7 arises primarily from its intrinsic biology (mechanism of action and downstream signaling), rather than from its structural stability. That said, improved stability and reduced aggregation propensity could potentially lower the immunogenicity risk of protein biologics. Nevertheless, there are currently no validated in vitro or in vivo assays that reliably correlate protein stability or aggregation with clinical immunogenicity outcomes.

(25) The acronym MSA needs to be defined.

We have defined the acronym MSA (Multiple Sequence Alignment) on page 7, line 142.

(26) The acronym CPD needs to be defined.

We have defined the acronym CPD (Computational Protein Design) on page 23, line 468.

**Reviewer #2 (Recommendations for the authors):**
Any experimental structural data would be good to have.

We plan to pursue X-ray crystallography of Neo-7 in future studies to obtain high-resolution structural confirmation. However, we emphasize that such experiments require significant time and resources, and the results would not alter the biological claims made in this study. Our focus here is to demonstrate that with recent advances in in silico protein structure prediction algorithms, such as AlphaFold2, it is now feasible to redesign therapeutic proteins with sufficient accuracy to achieve improved developability and biological performance. This study highlights how computational approaches can streamline protein drug engineering, reducing reliance on labor-intensive structural studies during the early stages of therapeutic development.

Please add details of how the changed kinetics might affect downstream pathways.

We appreciate the reviewer’s suggestion to elaborate on the biological implications of the altered binding kinetics.

Our data show that Neo-7 variants display a slower off-rate relative to WT-IL-7, which likely reflects enhanced stabilization of the cytokine–receptor complex. In principle, this could prolong receptor occupancy and modestly extend downstream signaling duration. However, several biological features of IL-7 constrain the risk of excessive or aberrant signaling:

Receptor Regulation: IL-7 signaling induces rapid downregulation of IL7Rα on T cells, serving as a feedback mechanism to prevent sustained or uncontrolled activation. This "hardwired" receptor regulation reduces the likelihood that a slower off-rate translates into pathological over-signaling.

Pathway Specificity: IL-7 primarily signals through the JAK/STAT5 axis, with little evidence of signaling bias. Unlike other cytokines (e.g., IL-21, IL-22) that can activate STAT1 or STAT3 and drive distinct functional outcomes, IL-7’s pathway specificity minimizes concerns about altered signaling directionality.

Transcriptional Evidence: Our RNA-seq analysis further supports this, showing that Neo-7 and WT-IL-7 activate similar transcriptional programs. The differences we observed were in the magnitude of response, not in the qualitative nature of the pathways engaged. This suggests that Neo-7 variants enhance the intensity of canonical IL-7 signaling rather than redirecting it toward alternative or unintended pathways.

Together, these findings support the interpretation that the slower off-rate of Neo-7 variants likely contributes to stronger or more sustained activation of IL-7’s canonical STAT5 pathway, while intrinsic regulatory mechanisms and pathway fidelity safeguard against inappropriate signaling outcomes.

Minor:(1) The Figure 3 text is hard to read.

We acknowledge the reviewer’s concern regarding the readability of Figure 3. In the revised manuscript, we will provide a higher-resolution version of the figure to ensure that all labels and text are clearly visible upon magnification.

(2) The manuscript switches between "Neo-7" and "Neo7" .

We agree with the reviewer’s observation. To maintain consistency throughout the manuscript, all references have been standardized to Neo-7.